# Learning with Instance-Dependent Label Noise: Maintaining Accuracy and Fairness

## Abstract

Incorrect labels hurt model performance when the model overfits to noise. Many state-of-the-art approaches that address label noise assume that label noise is independent from the input features. In practice, however, label noise is often feature or instance-*dependent*, and therefore is biased (i.e., some instances are more likely to be mislabeled than others). Approaches that ignore this dependence can produce models with poor discriminative performance, and depending on the task, can exacerbate issues around fairness. In light of these limitations, we propose a two-stage approach to learn from datasets with instance-dependent label noise. Our approach utilizes alignment points, a small subset of data for which we know the observed and ground truth labels. On many tasks, our approach leads to consistent improvements over the state-of-the-art in discriminative performance (AUROC) while maintaining model fairness (area under the equalized odds curve, AUEOC). For example, when predicting acute respiratory failure onset on the MIMIC-III dataset, the harmonic mean of the AUROC and AUEOC of our approach is 0.84 (SD 0.01) while that of the next best baseline is 0.81 (SD 0.01). Overall, our approach leads to learning more accurate and fair models compared to existing approaches in the presence of instance-dependent label noise.

## 1 Introduction

Datasets used to train machine learning models can contain incorrect labels (i.e., label noise). While label noise is widely studied, the majority of past work focuses on when the noise is independent from an instance's features (i.e., instance-independent label noise) [Song et al. (2020)]. However, label noise is sometimes biased and depends on an instance's features (i.e., instance-dependent) [Wei et al. (2022b)], leading to different noise rates within subsets of the data. This results in model overfitting, and in tasks where the dataset contains instances from different groups corresponding to some sensitive attribute, this can also lead to disparities in performance [Liu (2021)]. For example, consider the task of predicting cardiovascular disease among patients admitted to a hospital. Compared to male patients, female patients may be more likely to be misdiagnosed [Maserejian et al. (2009)] and thus mislabeled, potentially leading to worse predictions for female patients. Although instance-dependent label noise has recently received more attention [Cheng et al. (2020b); Xia et al. (2020); Wang et al. (2021a)], the effect of these approaches on model fairness has been relatively understudied [Liu (2021)]. Here, we address the limitations of current approaches and propose a novel method for learning with instance-dependent label noise, specifically examining how modeling assumptions affect existing issues around model fairness.

Broadly, current work addressing instance-dependent label noise falls into one of two categories: 1) that which learns to identify mislabeled instances [Cheng et al. (2020a); Xia et al. (2022); Zhu et al. (2022a)], and 2) that which learns to optimize a noise-robust objective function [Feng et al. (2020); Wei et al. (2022a)]. In the first category, instances identified as mislabeled are either filtered out [Kim et al. (2021)] or relabeled [Berthon et al. (2021)]. In some settings, this approach can have a negative effect on model fairness. For example, when instances represent individuals belonging to subgroups defined by a sensitive attribute, approaches that filter out mislabeled individuals could ignore a disproportionately higher number of individuals from subgroups with more label noise. While relabeling approaches use all available data, they can be sensitive to assumptions around the noise distribution [Ladouceur et al. (2007)]. In the second category, current approaches rely on objective functions that are less prone to overfitting to the noise, while using all of the data and

Table 1: Notation. We summarize our notation, with the notation appearing in the left column and a description in the right column. Superscripts in parentheses represent specific instances (e.g., $\mathbf{x}^{(i)}$). Subscripts represents indexes into a vector (e.g., $\mathbf{x}_i$)

| Notation | Description | Notation | Description |
|---|---|---|---|
| n | Number of instances | d | Number of features |
| c | Number of classes | G | Number of groups |
| $\mathbf{x} \in \mathbb{R}^d$ | Feature vector | $\tilde{y} \in \{1, 2, ..., c\}$ | Observed label |
| $\hat{y} \in [0, 1]^c$ | Predicted class probabilities | $y \in \{1, 2, ..., c\}$ | Ground truth label |
| $\hat{\beta}_\phi$ | $P(y == \tilde{y}|\tilde{y}, \mathbf{x})$ | A | Set of alignment points |
| $\theta$ | Main model parameters | $\phi$ | Auxiliary model parameters |

observed labels [Chen et al. (2021)]. However, like the first category, many rely on assumptions like the memorization effect, and thus, potentially suffer from the same limitations [Wang et al. (2021a)].

In light of these limitations, we propose an approach to address instance-dependent label noise, makes no assumptions about the noise distribution, and uses all data during training. We leverage a set of representative points in which we have access to both the observed and ground truth labels. While past work has used the observed labels as the ground truth labels for anchor points [Xia et al. (2019); Wu et al. (2021)], we consider a different setting in which the ground truth and observed labels do not agree for some points. To make the differentiation clear, we refer to these points as 'alignment points'. Such a setting arises frequently in healthcare. Oftentimes, one labels an entire dataset using a proxy function (obtaining observed labels) but also labels a small subset of the data using manual review (obtaining ground truth labels). We use them to initialize the model's decision boundary and learn the underlying pattern of label noise in pre-training. We then add the remaining data for fine-tuning, minimizing a weighted cross-entropy loss based on the learned noise pattern.

On synthetic and real data, we evaluate our approach in terms of discriminative performance and model fairness, measured using the area under the receiver operator curve (AUROC) and area under the equalized odds curve (AUEOC). We demonstrate that our approach improves on state-of-the-art baselines from the noisy labels and fairness literature, such as stochastic label noise [Chen et al. (2021)] and group-based peer loss [Wang et al. (2021b)]. Overall, our contributions are: 1) a novel approach to learn from datasets with instance-dependent noise; 2) a systematic examination of different settings of label noise, showing where approaches fail with respect to discriminative performance and fairness; 3) empirical results showing that the proposed approach is robust to both to the noise rate and amount of noise disparity between subgroups, reporting the model's ability to maintain discriminative performance and fairness; 4) a demonstration of how performance of the proposed approach changes when assumptions about the alignment set are violated.

## 2 METHODS

We introduce a two-stage approach for learning with instance-dependent label noise that leverages a small set of alignment points for which we have both observed and ground truth labels.

**Notation and Problem Setup** Our main notation is in **Table 1**. We train a model using dataset $D = \{\mathbf{x}^{(i)}, \tilde{y}^{(i)}\}_{i=1}^n$, where $\mathbf{x} \in R^d$ and $y \in \{1, 2, ..., c\}$ to learn a function $f : \mathbf{x} \to y$ that can map unseen instances into one of $c$ classes based on their feature vectors. In the presence of noisy labels, $\tilde{y}$ is not always equal to $y$. For each alignment point, we know both $\tilde{y}$ and $y$. This is in contrast to the rest of the data, where we only know $\tilde{y}$. We aim to learn model parameters, $\theta$, such that $\theta(\mathbf{x})$ represents the predicted class probabilities, (i.e., $\hat{y}$). Alignment points are similar to anchor points [Xia et al. (2019)], but we do not assume that $\tilde{y}^{(i)} = y^{(i)}$ for these points.

For the rest of the paper, we focus on the following case of instance-dependent noise. Let $f$ be the function used to generate the ground truth labels (i.e., $f(\mathbf{x}) = y$), and let $\tilde{m}$ be the function used to generate an instance's risk of being mislabeled (i.e., $y = \tilde{y}$ if $\tilde{m}(\mathbf{x})$ is above some threshold). In the following toy example, suppose $f(\mathbf{x}) = 1$ for true positive instances, and $\tilde{m}(\mathbf{x}) = 1$ if $x_1 > 0.5$ and 0 otherwise. Here, instances where $x_1 > 0.5$ have noisy labels. Although $f$ and $\tilde{m}$ were deterministic for simplicity, they can also be based on probabilistic functions. We denote the set of alignment points (i.e., the alignment set) as set $A$ consisting of instances $i$ such that both $\tilde{y}^{(i)}$

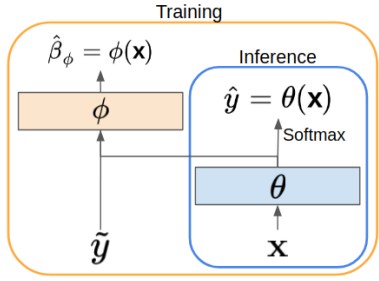

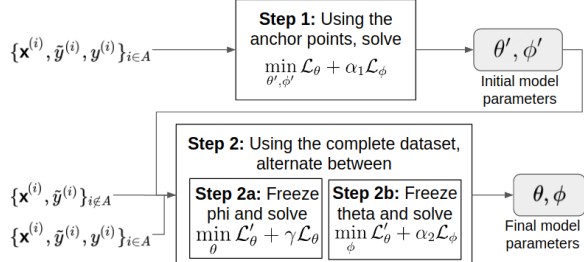

(a) Proposed network. The model makes class predictions, $\hat{y}$, at training and inference time using $\theta$. At training time, it also predicts whether the observed label is correct using $\phi$. $\theta$ and $\phi$ are pre-trained using $A$ and then fine tuned with the complete dataset.

(b) Proposed training pipeline. We first pretrain the model using the alignment points (i.e., data with observed and ground truth labels), then train on the noisy data. $\mathcal{L}_\theta$ is the objective to train on the alignment points. $\mathcal{L}_\phi$ is the objective to learn the label confidence score ($\hat{\beta}_\phi$). $\mathcal{L}'_\theta$ is the objective to train the noisy data. $\alpha_1$, $\alpha_2$, and $\gamma$ are hyperparameters.

Figure 1: Proposed Approach

and $y^{(i)}$ are known and assuming that it includes instances where $\tilde{y}^{(i)} \neq y^{(i)}$. Although it is not always possible to obtain alignment points, the setting above does occur in practice. For example, in healthcare, automated labeling tools based on the structured contents of the electronic health record are often applied to identify cohorts or outcomes of interest [Upadhyaya et al. (2017); Norton et al. (2019); Tjandra et al. (2020); Yoo et al. (2020)]. However, it is often overlooked that such practical definitions are not always reflective of ground truth, and thus, require validation by comparing to a subset of the data that was chart reviewed by an expert. This subset can be derived from a randomly chosen individuals, and thus can be constructed to be representative of the target population.

**Proposed Approach** We describe the proposed network and training procedure below.

Proposed Network. Our proposed network (**Figure 1a**) consists of two components. The first, $\theta$, is a feed-forward network that uses feature vector $\mathbf{x}$ to predict the class probabilities, $\hat{y} = \theta(\mathbf{x})$. The second component, $\phi$, is an auxiliary feed-forward network that uses observed label $\tilde{y}$ and the pre-softmax output of $\theta$ to compute $\hat{\beta}_\phi = P(y == \tilde{y}|\tilde{y}, \mathbf{x})$, an instance-dependent prediction for whether the observed label is correct based on $\mathbf{x}$ and $\tilde{y}$. $\hat{\beta}_\phi$ can be considered as a confidence score for the observed label, with higher values indicating higher confidence. Learning $\hat{\beta}_\phi$ learns the underlying pattern of label noise by forcing the model to distinguish between which examples are correctly or incorrectly labeled. This will be helpful during training, which we describe below. By including the observed label as input to $\phi$, our approach also applies to instance-independent label noise by accounting for the case when the underlying pattern of label noise cannot be learned from the features alone. At training time, we compute the loss using the outputs from both $\theta$ and $\phi$. At inference time, we compute the class predictions from $\theta$ only since $\tilde{y}$ is not available.

Training Procedure. Our procedure is summarized in **Figure 1b** and **Appendix A**. In Step 1, we pre-train the $\theta$ and $\phi$ networks using the alignment points, minimizing an objective function based on cross entropy: $\min_{\theta',\phi'} \mathcal{L}_\theta + \alpha_1 \mathcal{L}_\phi$. $\alpha_1 \in \mathbb{R}^+$ is a scalar hyperparameter; $\theta'$ and $\phi'$ are the initial parameters of $\theta$ and $\phi$. $\mathcal{L}_\theta$, is the cross entropy loss between the class predictions and ground truth labels, learning the weights for $\theta$, and thus, the model's decision boundary. $\mathbb{I}$ is an indicator function.

$$\mathcal{L}_\theta = \frac{-1}{|A|} \sum_{i \in A} \sum_{j=1}^{c} \mathbb{I}\left(y^{(i)} == j\right) log\left(\hat{y}_j^{(i)}\right)$$

$\mathcal{L}_\phi$ is the cross entropy loss between the predicted confidence score $\hat{\beta}_\phi^{(i)}$ and the actual agreement between $\tilde{y}^{(i)}$ and $y^{(i)}$. It learns the weights for $\phi$, and in turn, the underlying label noise pattern.

$$\mathcal{L}_\phi = \frac{-1}{|A|} \sum_{i \in A} \mathbb{I}\left(\tilde{y}^{(i)} == y^{(i)}\right) log\left(\hat{\beta}_\phi^{(i)}\right) + \mathbb{I}\left(\tilde{y}^{(i)} \neq y^{(i)}\right) log\left(1 - \hat{\beta}_\phi^{(i)}\right)$$

In Step 2, we initialize $\theta$ and $\phi$ using $\theta'$ and $\phi'$ from Step 1 and fine tune them using the complete dataset, which potentially contains noisy labels. Step 2 can be further divided into two parts, Step

2a and Step 2b. Each part aims to improve a specific component of the network (e.g., $\theta$) using another component of the network (e.g., $\phi$). Throughout Step 2, we alternate between Step 2a and Step 2b in a manner similar to expectation maximization so that we continually improve both $\theta$ and $\phi$. In Step 2a, we freeze $\phi$ and train $\theta$ by minimizing the objective $\min_\theta \mathcal{L}'_\theta + \gamma \mathcal{L}_\theta$. $\gamma \in \mathbb{R}^+$ is a scalar hyperparameter. In Step 2b, we freeze $\theta$ and train $\phi$ using the objective $\min_\phi \mathcal{L}'_\theta + \alpha_2 \mathcal{L}_\phi$. $\alpha_2 \in \mathbb{R}^+$ is a scalar hyperparameter. $\mathcal{L}'_\theta$ computes the cross entropy loss over the potentially noisy, non-alignment points. Each instance is weighted by the model's confidence in whether the observed label is correct via $\hat{\beta}_\phi^{(i)}$, taking advantage of the model's learned noise pattern.

$$\mathcal{L}'_\theta = \frac{-1}{|\overline{A}|} \sum_{k=1}^G \frac{1}{1 - \hat{r}_k} \sum_{i \in \overline{A} \cap g_k} \sum_{j=1}^c \hat{\beta}_\phi^{(i)} \mathbb{I}\left(\tilde{y}^{(i)} == j\right) log\left(\hat{y}_j^{(i)}\right)$$

The quantity $1 - \hat{r}_k$ is as follows. We introduce sets $g_k$ for $k = 1, 2, ..., G$ to represent disjoint subgroups of interest in the data, where $g_a \cap g_b = \emptyset$ for all $a = 1, 2, ..., G$, $b = 1, 2, ..., G$ with $a \neq b$ and $\cup_{k=1}^G g_k = D$. Each group $g_k$ is then associated with estimated noise rate $\hat{r}_k = \frac{1}{|g_k|} \sum_{i \in g_k} 1 - \hat{\beta}_\phi^{(i)}$ and estimated clean (i.e., correct) rate $1 - \hat{r}_k = \frac{1}{|g_k|} \sum_{i \in g_k} \hat{\beta}_\phi^{(i)}$. Although weighting each instance by $\hat{\beta}_\phi$ is a form of soft filtering, weighting each group by the inverse of its overall clean rate avoids the effect of de-emphasizing groups with higher predicted noise rates. As a result, the expected value of $\mathcal{L}'_\theta$ with respect to $\hat{\beta}_\phi$ is equal to the cross entropy loss between the model's predictions and ground truth labels (see **Appendix A** for proof). However, this assumes accurate estimates of $\hat{\beta}_\phi$. Thus, we expect that the proposed approach will perform best when the alignment set is representative of the target population (i.e., there is no dataset shift from the alignment set to the remaining data), since this will produce the best estimate of $\hat{\beta}_\phi$. In **Section 4**, we test this.

Our approach encourages fairness, as measured by equalized odds [Hardt et al. (2016)], by upweighting groups with a higher estimated noise rate so that they are not dominated/ignored compared to groups with a lower estimated noise rate. In doing so, we hypothesize that accuracy and related metrics, such as the false positive rate, will improve, thereby improving equalized odds. We focus on equalized odds since, in the domains of interest, metrics like the true and false positive rates are particularly important. For example, in healthcare, we would like to correctly predict who will and will not develop a condition so that the appropriate treatment may be used.

During Step 2a, $\mathcal{L}'_\theta$ is used to train $\theta$ by learning to predict $\hat{y}$ such that it matches observed label $\tilde{y}$ on instances that are predicted to be correctly labeled. During Step 2b, $\mathcal{L}'_\theta$ is used to train $\phi$. Here, since $\theta$ is frozen and $\phi$ is not, the network learns to predict the optimal $\hat{\beta}_\phi$. Based on $\mathcal{L}'_\theta$ alone, there are two possible ways to learn $\hat{\beta}_\phi$. The first is to consistently make $\hat{\beta}_\phi$ close to 0. The second is to learn $\hat{\beta}_\phi$ such that it is close to 1 when $\hat{y}$ matches $\tilde{y}$ and close to 0 when $\hat{y}$ does not match $\tilde{y}$. Since $\tilde{y}$ is used as a proxy for $y$ in this step, the second way aligns with what we want $\hat{\beta}_\phi$ to represent. To encourage this over the first way (i.e., consistently predicting 0 for $\hat{\beta}_\phi$), we include $\mathcal{L}_\phi$ in Step 2b, which is *not* minimized by consistently predicting 0 for $\hat{\beta}_\phi$. Note that, in order to expect a benefit by including the noisy data in Step 2b, we rely on the cluster assumption [Singh et al. (2008)] from semi-supervised learning, which broadly states that labeled data fall into clusters and that unlabeled data aid in defining these clusters. In the context of Step 2b, 'labeled' and 'unlabeled' are analogous to whether we know if the ground truth and observed labels match (i.e., alignment point versus non-alignment point), rather than the actual class labels themselves. As a result, we rely on the alignment set being representative of the target population here as well to avoid dataset shift.

In contrast to previous filtering approaches, our approach utilizes all data during training. Moreover, it does not require a specialized architecture beyond the auxiliary network to compute $\hat{\beta}_\phi$. Thus, it can be used to augment existing architectures in a straightforward manner.

## 3 EXPERIMENTAL SETUP

We empirically explore the performance of our proposed approach relative to state-of-the-art baselines on five benchmark prediction tasks with two different label noise settings. Implementation details are provided in **Appendices C** and **D**.

**Datasets** We consider five different binary prediction tasks on four datasets from several domains on synthetic and real datasets. While work in noisy-label learning often focuses on image datasets (e.g., MNIST or CIFAR 10/100), given our emphasis on fairness, we selected datasets reflective of tasks in which harmful biases can arise (e.g., predicting clinical outcomes, recidivism, and income).

Synthetic: We generated a dataset containing 5,000 instances as described below. Each instance had 30 features, and features were drawn from a standard normal distribution. We defined the feature at index 0 to be a synthetic sensitive attribute. Instances with values below the $20^{th}$ percentile for this feature were considered as the 'minority', and the rest were considered as the 'majority'. Features 10-19 for the majority instances and features 20-29 for the minority instances were set to 0 to provide more contrast between the two groups. Coefficients describing the contribution of each feature to the outcome, $\mathbf{w}$, were also drawn from a standard normal distribution, and an outcome $z$ was computed by taking the dot product of the features and coefficients and passing it through a sigmoid. A binary ground truth label, $y$, for each instance was assigned, based on $z$. Instances whose value for $z$ fell above the $50^{th}$ percentile were given the label 1. All other instances were labeled 0. The positive rates for the synthetic majority and minority groups were 37.5% and 32.3%, respectively.

MIMIC-III: Within the healthcare domain, we leverage a publicly available dataset of electronic health record data [Johnson et al. (2016)]. We considered two separate prediction tasks: onset of 1) acute respiratory failure (ARF) and 2) shock in the ICU (intensive care unit) [Oh et al. (2019)]. MIMIC-III includes data pertaining to vital signs, medications, diagnostic and procedure codes, and laboratory measurements. We considered the four hour prediction setup for both tasks as described by [Tang et al. (2020)]. This resulted in 15,873 and 19,342 ICU encounters for each task, respectively. After preprocessing (see **Appendix B**), each encounter had 16,278 and 18,186 features, for each task respectively. We used race as a sensitive attribute, with about 70% of patients being White (positive rate 4.5% [ARF], 4.1% [shock]) and 30% being non-White (4.4% [ARF], 3.7% [shock]).

Adult: a publicly available dataset of census data [Dua & Graf (2017)]. We considered the task of predicting whether an individual's income is over $50,000. This dataset includes data pertaining to age, education, work type, work sector, race, sex, marital status, and country. Its training and test sets contain 32,561 and 16,281 individuals, respectively. We used a pre-processed version of this dataset[1]. We only used 1,000 randomly selected individuals out of 32,561 during training and only included features pertaining to age, education, work type, marital status, work sector, and sex to make the task more difficult (see **Appendix B**). We also normalized each feature to have a range of 0-1. This resulted in 56 features per individual. We used sex as a sensitive attribute, with 67.5% of individuals being male (positive rate 30.9%) and 32.5% being female (positive rate 11.3%).

COMPAS: a publicly available dataset collected by ProPublica from Broward County, Florida, USA [Angwin et al. (2016)]. We considered the task of predicting recidivism within two years, i.e., whether a criminal defendant is likely to re-offend. COMPAS includes data pertaining to age, race, sex, and criminal history. We used a pre-processed version of this dataset[2] and also normalized each feature to have a range of 0-1 (see **Appendix B**). This resulted in 6,172 individuals with 11 features per individual. We used race as a sensitive attribute, with 65.8% of individuals being White (positive rate 39.1%) and 34.2% being non-White (positive rate 44.5%).

**Label Noise** To test the robustness of our approach in different settings of label noise, we introduce noise in two ways. Like past work [Song et al. (2020)], our setup is limited in that our noise is synthetic and we use the labels in the real datasets as ground truth, since we do not have access to actual ground truth labels on these public datasets. For uniform random noise, each instance had its label flipped with a fixed probability (e.g., 0.3). The noise rates were the same between subgroups. For instance-dependent noise, mislabeling was a function of the features. Let $\mathbf{w}_m$ ($\mathbf{w}_m \sim N(0, 0.33)^D$) and $z_m$ ($z_m = \sigma(\mathbf{x} \cdot \mathbf{w}_m)$) denote the coefficients describing the contribution of each feature to mislabeling and the risk of mislabeling, respectively. Whether an instance was mislabeled was based on $z_m$ and the desired noise rate ($\tilde{m}(\mathbf{x}) = y \ if \ z_m > 1 - noise \ rate, \ 1 - y \ otherwise$). For example, for a noise rate of 30%, instances whose value for $z_m$ was above the $70^{th}$ percentile had their labels flipped. This allowed us to vary the noise rate within subgroups in a straightforward manner. Across datasets, we focused on cases where the noise rate in the 'minority' population was always greater than or equal to that of the 'majority' group since this is more likely to occur.

---

[1]https://github.com/AissatouPaye/Fairness-in-Classification-and-Representation-Learning
[2]https://www.kaggle.com/danofer/compass

**Evaluation Metrics** We evaluate our proposed approach in terms of discriminative performance and model fairness. For discriminative performance, we evaluate using the area under the receiver operating characteristic curve (AUROC) (higher is better). For model fairness, we measure the area under the equalized odds curve (AUEOC) [de Freitas Pereira & Marcel (2020)] (higher is better). We use the AUEOC rather than equalized odds to avoid choosing a classification threshold to measure the true and false positive rates. For classification threshold $\tau$, we calculate the equalized odds (EO($\tau$)) between two groups, called 1 and 2, as shown below. $TP_a(\tau)$ and $FP_a(\tau)$ denote true and false positive rates for group $a$ at threshold $\tau$, respectively. The AUEOC is obtained by plotting the EO against all possible values of $\tau$ and calculating the area under the curve. We also compute the harmonic mean (HM) between the AUROC and AUEOC to highlight how the approaches maintained discriminative performance and fairness. We compute the harmonic mean so that the overall measure is more heavily weighted by the worse performing metric. For example, if a classifier has AUROC=0.5 and AUEOC=1.0, we emphasize the poor discriminative performance.

$$EO(\tau) = \frac{2 - |TP_1(\tau) - TP_2(\tau)| - |FP_1(\tau) - FP_2(\tau)|}{2}$$

**Baselines** We evaluate our proposed approach in the context of several baselines to test different hypotheses. Standard does not account for label noise and assumes that $\tilde{y} = y$ is always true. SLN + Filter [Chen et al. (2021)] combines filtering [Arpit et al. (2017)] and SLN [Chen et al. (2021)] and was shown to outperform state-of-the-art approaches such as Co-Teaching [Han et al. (2018)] and DivideMix [Li et al. (2020)]. It relies on filtering heuristics, which indirectly rely on uniform random label noise to maintain discriminative performance and model fairness. JS (Jensen-Shannon) Loss [Englesson & Azizpour (2021)] builds from semi-supervised learning and encourages model consistency when predicting on perturbations of the input features and was shown to be competitive with other state-of-the-art noise-robust loss functions [Ma et al. (2020)]. It was proposed in the context of instance-independent label noise. Transition [Xia et al. (2020)] learns to correct for noisy labels by learning a transition function and was shown outperform state-of-the-art approaches such as MentorNet [Jiang et al. (2018)]. It applies to instance-dependent label noise, but it assumes that the contributions of each feature to mislabeling and input reconstruction are identical. CSIDN (confidence-scored instance-dependent noise) [Berthon et al. (2021)] also learns a transition function and was shown outperform state-of-the-art approaches such as forward correction [Patrini et al. (2017)]. Like our approach, CSIDN uses the concept of 'confidence' in the observed label to help with training. Unlike our approach, CSIDN uses the model's class predictions directly as confidence scores (instead predicting them via an auxiliary network) and uses them to learn the transition function (as opposed to re-weighting the loss). Fair GPL [Wang et al. (2021b)] builds on work addressing uniform random label noise [Jiang & Nachum (2020)] and uses peer loss (i.e., data augmentation that reduces the correlation between the observed label and model's predictions) within subgroups defined by a sensitive attribute [Wang et al. (2021b)]. It assumes that label noise depends on only the sensitive attribute. We also train a model using the ground truth labels (referred to as Clean Labels) as an empirical upper bound for discriminative performance.

## 4 Results and Discussion

We describe the results from experiments with instance-dependent noise. For each plot, we combined discriminative performance and fairness and plotted the HM of the AUROC and AUEOC to assess general performance with respect to both of these aspects.

**Is the Approach Robust to Varying Noise Levels?** Here, we investigated how robust the proposed approach and baselines were to varying amounts of instance-dependent label noise (**Figure 2a**). On the synthetic dataset, we considered noise rates between 10% and 60% in the majority group. For the minority group, we considered noise rates of that were consistently 20% higher than that of the majority group, to keep the noise disparity level (i.e., the difference in noise rates between subgroups) constant. As expected, all approaches degrade as the noise rate increases, with the proposed approach experiencing the least degradation up to a majority noise rate of 60%. The baseline Transition was also robust to the noise rate, although overall performance was generally worse than the proposed approach. At a noise rate of 60%, it performed similarly to the proposed approach. The other approaches experienced more degradation, likely because they were for instance-independent label noise, highlighting the need for approaches that are designed for instance-dependent noise.

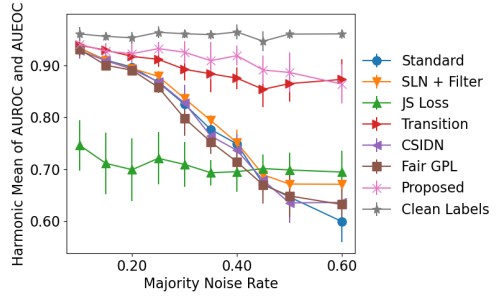 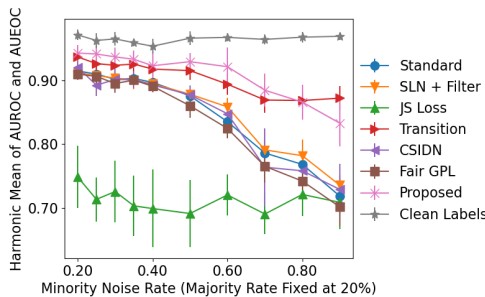

(a) As the noise rate increases, the proposed approach generally shows the least degradation.

(b) As the noise disparity increases, the proposed approach generally shows the least degradation.

Figure 2: Robustness to noise rate and noise disparity in an instance dependent setting. We plot the mean and standard deviation for 10 random seeds.

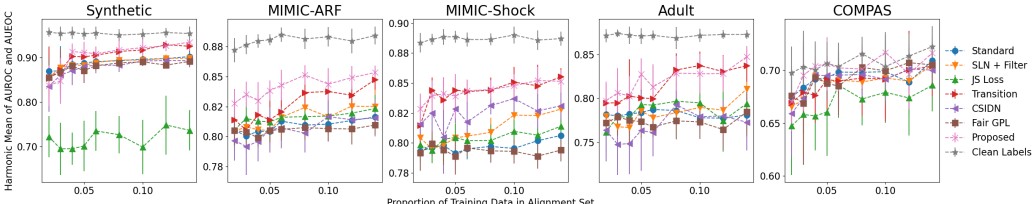

(a) As we decrease the alignment set size (proportion of training data) performance decreases. Still, at an alignment set size of 5%, the proposed approach generally outperforms the baselines.

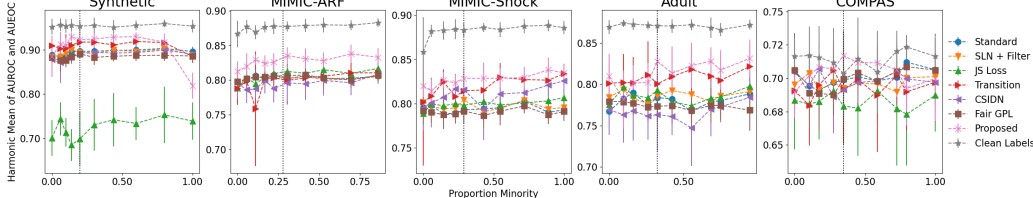

(b) As we vary the alignment set bias (proportion of minority instances) performance varies. The proposed approach is generally robust to changes in the bias of the alignment set. The dashed vertical black line shows the proportion at which the minority group occurs in the dataset (i.e., an unbiased alignment set).

Figure 3: Robustness to varying alignment sets. Mean and standard deviation for 10 random seeds.

**Is the Approach Robust to Varying Noise Disparities?** Here we investigated how robust the proposed approach and baselines were to varying levels of noise disparity (**Figure 2b**). With a majority noise rate of 20% throughout, we evaluated the approaches with minority noise rates between 20% and 90% (i.e., from no disparity to a disparity of 70%) on the synthetic dataset. In line with our expectations, the proposed approach was robust to changes in the noise disparity. This is likely because the objective function $\mathcal{L}'_\theta$ from Step 2 of training accounts for disparities by scaling each instance-specific loss term with the reciprocal of its estimated group clean rate (i.e., 1 - the group noise rate). Most of the baselines, such as SLN + Filter, which learn to filter out noisy instances, experienced a larger drop in performance as the disparity increased. This aligns with our hypothesis. Although the baseline Transition also demonstrated robustness to the noise disparity and outperformed the proposed approach at minority noise rates higher than 80%, this setting is unlikely to be realistic.

**Does Alignment Set Composition Matter?** Our next set of experiments tests the proposed approach in settings where we relax key assumptions about the alignment set. We consider all datasets with instance-dependent noise. The majority/minority noise rates were 20%/40%, respectively.

Part 1: Alignment set size. We varied the size of the alignment set, testing at sizes between 1% and 15% of the training set, with the alignment set representative of the test set (**Figure 3a**). As expected, when the alignment set is small, performance worsens. However, the proposed method generally

improves over the baselines at an alignment set size of as small as 3% of the training data. On the real datasets, the proposed approach had a higher HM compared to the baselines at an alignment set size of $< 5\%$, showing improved overall performance. As the size of the alignment set increased, some baseline approaches were able to match that of the proposed method (e.g., for MIMIC-Shock, the baseline Transition had similar performance to the proposed method at an alignment set $> 5\%$). This is in line with expectations, as this makes the overall training set cleaner.

Part 2: Biased alignment set. We hypothesized that our approach works best when the alignment points are representative of the population. Here, we test how the proposed approach performs when this assumption is violated. We varied the amount of bias in the alignment set by changing the proportion at which the subgroups were present. We kept the size of the alignment set constant at 10% of the training data (2% for MIMIC-III on both tasks). We observe that the proposed approach is beneficial over a wide range of conditions, i.e., when the minority proportion is 20%-80% (**Figure 3b**). We hypothesize that this could be because the learned relationship between the features and noise can generalize across groups to an extent. In scenarios where the proposed approach is less beneficial compared to the baselines, one subgroup heavily dominated the alignment set. This is shown in **Figure 3b** on the extremes of the x-axis, which correspond to an alignment set that is heavily over-represented for one subgroup and heavily under-represented for the other. Our approach relied, in part, on having a relatively unbiased alignment set for estimating $\hat{\beta}_\phi$ in order to avoid introducing dataset shift between the two steps of our training pipeline. Thus, these results are in line with our expectations and highlight a limitation of our approach. However, despite this reliance, we observe that the our approach is still robust in some scenarios where the alignment set is biased.

**Additional Experiments**: We include results for experiments where we vary the proportion at which the minority group appears in the entire dataset, results for uniform random noise, an ablation study, and a hyperparamter sensitivity analysis in **Appendix E**.

## 5 RELATED WORK

Label noise is a well-studied problem in machine learning [Song et al. (2020)] and may be instance-independent or -dependent. Regardless of the setting, past approaches fall into two broad categories. The first aims to identify mislabeled instances via filtering heuristics or by estimating the probability of mislabeling [Patrini et al. (2017); Li et al. (2020); Cheng et al. (2020b)]. The second adapts regularization techniques, such as label smoothing and contrastive learning to noisy labels [Manwani & Sastry (2013); Menon et al. (2019); Xue et al. (2022); Liu et al. (2022)]. We also examine how label noise has been tackled in the fairness literature and how past work has used anchor points.

**Identifying Mislabeled Data** Approaches that learn to identify mislabeled instances fall into two sub-categories: 1) filtering approaches and 2) relabeling approaches. Filtering approaches use heuristics to identify mislabeled instances (e.g., MentorNet [Jiang et al. (2018)], Co-teaching [Han et al. (2018)], FINE [Kim et al. (2021)]). Many are based on the idea that correctly labeled instances are easier to classify than mislabeled instances (i.e., the memorization effect) [Arpit et al. (2017)]. For example, mislabeled instances could be those that the model incorrectly classifies [Verbaeten (2002); Khoshgoftaar & Rebours (2004); Thongkam et al. (2008); Chen et al. (2019)], have a high loss value [Yao et al. (2020a)], or significantly increase the complexity of the model [Gamberger et al. (1996)]. Given the identified mislabeled instances, these approaches either ignore them during training [Zhang et al. (2020)] or treat them as 'unlabeled' and apply techniques from semi-supervised learning (e.g., DivideMix [Li et al. (2020)], SELF [Nguyen et al. (2020)]). Overall, these heuristics have been shown to improve discriminative performance. However, depending on the setting, they can disproportionately discard subsets of data, which could hurt model fairness.

For binary classification, some approaches 'correct' (i.e., switch) the observed label for such instances [Han et al. (2020); Zheng et al. (2020)]. Building on this idea, others make use of a transition function that estimates the probability of the observed label being correct. Model predictions can then be adjusted by applying the function to the classifier's predictions for each class. Some works manually construct the transition function from expert knowledge [Patrini et al. (2017)], while others learn it [Xiao et al. (2015); Xu et al. (2019); Yao et al. (2020b); Xia et al. (2020); Zheng et al. (2021); Berthon et al. (2021); Jiang et al. (2022); Bae et al. (2022)]. However, such approaches often make assumptions on the form of the noise distribution, and past work has shown that results are sensitive to the choice of distribution [Ladouceur et al. (2007)].

To date, much of the work described above assumes instance-independent label noise (i.e., mislabeling is independent of the features). However, when this assumption is violated, the model may overfit to label noise [Lukasik et al. (2020)]. From an emerging body of work in instance-dependent label noise [Cheng et al. (2020b); Xia et al. (2020); Wang et al. (2021c); Zhu et al. (2022b)], current approaches remain limited in that they still rely on filtering heuristics. Although we use soft filtering, we filter based on the learned relationship between the features and noise rather than existing heuristics and upweight groups with a higher estimated noise rate. While similar to a transition function in some aspects, our approach requires fewer probability estimates on label correctness (two estimates compared to $c^2$ for a transition function) while achieving state-of-the-art performance.

**Noise-Robust Loss Functions** Prior work examines how regularization techniques can be adapted to the noisy labels setting, addressing issues related to overfitting on noisy data [Menon et al. (2019); Lukasik et al. (2020); Englesson & Azizpour (2021)]. Recently, label smoothing, and in some cases negative label smoothing, were found to improve the accuracy on both correctly labeled and mislabeled data [Lukasik et al. (2020); Wei et al. (2022a)]. With this approach, the observed labels are perturbed by a small, pre-determined value, with all labels receiving the same perturbation at every training epoch. Follow-up work found that, instead of applying the same perturbation at each epoch, adding a small amount of Gaussian stochastic label noise (SLN) at each epoch resulted in further improvements, as it helped to escape from local optima [Chen et al. (2021)]. However, these approaches were most beneficial in the context of augmenting existing methods that identify mislabeled instances (e.g., stochastic label noise is applied to instances that are identified as correctly labeled by filtering approaches), and thus, potentially suffer from the same limitations. Alternatively, recent work has also proposed perturbing the features instead of the labels to encourage consistency in the model's predictions [Englesson & Azizpour (2021)], though mainly in the context of instance-independent label noise. Others have proposed noise-robust variations of cross entropy loss [Feng et al. (2020); Wang et al. (2021a)] that generally relied on assumptions like the memorization effect.

**Label Noise and Fairness** Label noise has also been addressed within the fairness literature recently. When the frequencies at which subgroups (defined by a sensitive attribute) appear are different within a dataset, past work has shown that common approaches addressing label noise can increase the prediction error for minority groups (i.e., rarer subgroups) [Liu (2021)]. Past work proposed to re-weight instances from subgroups during training where model performance is poorer [Jiang & Nachum (2020)] in the random noise setting. Others use peer loss [Liu & Guo (2020)] within subgroups [Wang et al. (2021b)] but assume that noise depends only on the sensitive attribute. We also train with a weighted loss, but weights are based on predicted label correctness rather than performance on the observed labels. Recently, Wu et al. (2022) addressed some of the gaps of past work by examining the instance-dependent case. Our proposed approach differs from theirs in that we do not require our features to be grouped into distinct categories, such as root and low level attributes.

**Anchor Points for Addressing Label Noise** Another related setting in past work uses anchor points. To date, anchor points are generally used to learn a transition function [Xia et al. (2019; 2020); Berthon et al. (2021)] or for label correction directly [Wu et al. (2021)]. We use a similar concept, alignment points, to 1) pre-train the model, and 2) predict label correctness. The first part builds from work in semi-supervised learning [Cascante-Bonilla et al. (2021)], which has shown improvements from pre-training on labeled data. The second part is similar to a transition function, but differs in that we use the correctness predictions to re-weight the loss rather than adjust the predictions. We also assume that, for some alignment points, the ground truth and observed labels do not match.

## 6 CONCLUSION

We introduced a novel approach for learning with instance-dependent noisy labels. Our two-stage approach uses the complete dataset and learns the relationship between the features and label noise using a small set of alignment points. On synthetic and real datasets, we showed that the proposed approach led to improvements over state-of-the-art baselines in maintaining discriminative performance and model fairness. Our approach is not without limitations. We demonstrated that the success of the approach depends, in part, on the composition of the alignment set. Additionally, we only examined one form of fairness. Nonetheless this work brings together and builds on key aspects from the instance-dependent noisy labels literature, highlighting issues related to fairness.

REPRODUCIBILITY STATEMENT

**Section 3** describes our experimental setup, with additional details on implementation in **Sections B**, **C**, and **D** of the **Appendix**. The code has been included as a supplemental file and will be publicly released upon publication.

ETHICS STATEMENT

We do not believe that our paper has any ethical concerns. However, past work has shown that machine learning models can enforce harmful systemic biases if they are not carefully implemented Obermeyer et al. (2019). As a result, it is important to consider not only overall model performance, but also how the outcome is measured (i.e., choice of labels).

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

# APPENDIX

## A  PROPOSED APPROACH: ADDITIONAL DETAILS

We provide additional details on our proposed approach, including a pseudocode description and a justification on the loss of the proposed objective function.

### A.1  GENERAL OVERVIEW

We summarize our approach with pseudocode below in **Algorithm 1**. We begin with the dataset and initial model parameters, and we aim to use the dataset to learn the final model paramters.

---

**Algorithm 1** Proposed approach for learning with instance-dependent label noise. A is the set of anchor points. $\theta'$ and $\phi'$ are the initial model parameters for the $\theta$ and $\phi$ networks. Here, 'stopping criteria' may refer to any stopping criteria, such as early stopping. The Freeze() function takes as input model parameters and freezes them, and the Unfreeze() function takes as input model parameters and unfreezes them.

---

**Input:** $\{\mathbf{x}^{(i)}, \tilde{y}^{(i)}, y^{(i)}\}_{i \in A}, \{\mathbf{x}^{(i)}, \tilde{y}^{(i)}\}_{i \notin A}, \theta', \phi'$
**Output:** $\theta, \phi$ (final model parameters)
**Hyperparameters:** Scalars $\alpha_1, \alpha_2, \gamma$

Train $(\{\mathbf{x}^{(i)}, \tilde{y}^{(i)}, y^{(i)}\}_{i \in A}, \{\mathbf{x}^{(i)}, \tilde{y}^{(i)}\}_{i \notin A}, \theta', \phi')$

1: **while** $\neg$ stopping criteria **do**  $\quad\triangleright$ Step 1
2: $\quad\quad \hat{y} = \theta'(\mathbf{x})$  $\quad\triangleright$ Predict label
3: $\quad\quad \hat{\beta}_\phi = \phi'(\mathbf{x}, \tilde{y})$  $\quad\triangleright$ Predict label confidence
4: $\quad\quad \mathcal{L}_\theta = \frac{-1}{|A|} \sum_{i \in A} \sum_{j=1}^c \mathbb{I}\left(y^{(i)} == j\right) log\left(\hat{y}_j^{(i)}\right)$
5: $\quad\quad \mathcal{L}_\phi = \frac{-1}{|A|} \sum_{i \in A} \mathbb{I}\left(\tilde{y}^{(i)} == y^{(i)}\right) log\left(\hat{\beta}_\phi^{(i)}\right) + \mathbb{I}\left(\tilde{y}^{(i)} \neq y^{(i)}\right) log\left(1 - \hat{\beta}_\phi^{(i)}\right)$
6: $\quad\quad$ Loss $= \mathcal{L}_\theta + \alpha_1 \mathcal{L}_\phi$
7: $\quad\quad$ Update model parameters
8: $\quad\quad$ Compute stopping criteria
9: $\theta, \phi \leftarrow \theta', \phi'$
10: Freeze($\phi$)
11: **while** $\neg$ stopping criteria **do**  $\quad\triangleright$ Step 2
12: $\quad\quad \hat{y} = \theta'(\mathbf{x})$
13: $\quad\quad \hat{\beta}_\phi = \phi'(\mathbf{x}, \tilde{y})$
14: $\quad\quad \mathcal{L}_\theta = \frac{-1}{|A|} \sum_{i \in A} \sum_{j=1}^c \mathbb{I}\left(y^{(i)} == j\right) log\left(\hat{y}_j^{(i)}\right)$
15: $\quad\quad \mathcal{L}_\phi = \frac{-1}{|A|} \sum_{i \in A} \mathbb{I}\left(\tilde{y}^{(i)} == y^{(i)}\right) log\left(\hat{\beta}_\phi^{(i)}\right) + \mathbb{I}\left(\tilde{y}^{(i)} \neq y^{(i)}\right) log\left(1 - \hat{\beta}_\phi^{(i)}\right)$
16: $\quad\quad \mathcal{L}'_\theta = \frac{-1}{|\overline{A}|} \sum_{k=1}^G \frac{1}{1 - \hat{r}_k} \sum_{i \in \overline{A} \cap g_k} \sum_{j=1}^c \hat{\beta}_\phi^{(i)} \mathbb{I}\left(\tilde{y}^{(i)} == j\right) log\left(\hat{y}_j^{(i)}\right)$  $\quad\triangleright$ Weighted loss
17: $\quad\quad$ **if** $\phi$ is frozen **then**  $\quad\triangleright$ Step 2a
18: $\quad\quad\quad$ Loss $= \mathcal{L}_{\theta'} + \gamma \mathcal{L}_\theta$
19: $\quad\quad\quad$ Unfreeze($\phi$)
20: $\quad\quad\quad$ Freeze($\theta$)
21: $\quad\quad$ **else**  $\quad\triangleright$ Step 2b
22: $\quad\quad\quad$ Loss $= \mathcal{L}_{\theta'} + \alpha_2 \mathcal{L}_\phi$
23: $\quad\quad\quad$ Unfreeze($\theta$)
24: $\quad\quad\quad$ Freeze($\phi$)
25: $\quad\quad$ Update model parameters
26: $\quad\quad$ Compute stopping criteria
$\quad\quad$ **return** $\theta, \phi$  $\quad\triangleright$ Final model parameters, but only $\theta$ is used as inference time

---

## A.2 PROPOSED AND CLEAN LABEL LOSS

Here, we show that minimizing the proposed loss $\mathcal{L}'_\theta$ from step 2 of the proposed method is equal to minimizing cross entropy on the clean labels in expectation.

$$\mathcal{L}'_\theta = \frac{-1}{|\overline{A}|} \sum_{k=1}^{G} \sum_{i \in \overline{A} \cap g_k} \frac{1}{1 - \hat{r}_k} \sum_{j=1}^{c} \hat{\beta}_\phi^{(i)} \mathbb{I}\left(\tilde{y}^{(i)} == j\right) log\left(\hat{y}_j^{(i)}\right)$$

$$\mathbb{E}\left[ \sum_{k=1}^{G} \sum_{i \in \overline{A} \cap g_k} \frac{1}{1 - \hat{r}_k} \sum_{j=1}^{c} \hat{\beta}_\phi^{(i)} \mathbb{I}\left(\tilde{y}^{(i)} == j\right) log\left(\hat{y}_j^{(i)}\right) \right]$$

$$= \sum_{k=1}^{G} \sum_{i \in \overline{A} \cap g_k} \frac{1}{1 - \hat{r}_k} \sum_{j=1}^{c} \mathbb{E}\left[ \hat{\beta}_\phi^{(i)} \mathbb{I}\left(\tilde{y}^{(i)} == j\right) log\left(\hat{y}_j^{(i)}\right) \right]$$

$$= \sum_{k=1}^{G} \sum_{i \in \overline{A} \cap g_k} \frac{1}{1 - \hat{r}_k} \sum_{j=1}^{c} (1 - \hat{r}_k) \mathbb{I}\left(y^{(i)} == j\right) log\left(\hat{y}_j^{(i)}\right)$$

$$= \sum_{k=1}^{G} \sum_{i \in \overline{A} \cap g_k} \sum_{j=1}^{c} \mathbb{I}\left(y^{(i)} == j\right) log\left(\hat{y}_j^{(i)}\right)$$

As a reminder, each group $g_k$ is then associated with estimated noise rate $\hat{r}_k = \frac{1}{|g_k|} \sum_{i \in g_k} 1 - \hat{\beta}_\phi^{(i)}$ and estimated clean (i.e., correct) rate $1 - \hat{r}_k = \frac{1}{|g_k|} \sum_{i \in g_k} \hat{\beta}_\phi^{(i)}$. We can express the noise and clean rates in terms of $\hat{\beta}_\phi^{(i)}$ since

$$1 - r_k = \frac{1}{|g_k|} \sum_{i \in g_k} \mathbb{I}\left(\tilde{y}^{(i)} == y^{(i)}\right)$$

$$= P(y == \tilde{y}|\tilde{y}, \mathbf{x}) \; for \; a \; randomly \; chosen \; instance \; in \; g_k$$

$$= \frac{1}{|g_k|} \sum_{i \in g_k} P(y^{(i)} == \tilde{y}^{(i)}|\tilde{y}^{(i)}, \mathbf{x}^{(i)})$$

where $r_k$ and $1 - r_k$ are the actual noise and clean rates within group $k$, respectively. Therefore, since $\hat{\beta}_\phi$ is trained to predict $P(y == \tilde{y}|\tilde{y}, \mathbf{x})$, we estimate the noise and clean rates using $\hat{\beta}_\phi$.

## B PREPROCESSING DETAILS

Here, we provide more detail on real dataset pre-processing.

### B.1 MIMIC-III

Data were processed using the FlexIble Data Driven pipeLinE (FIDDLE), [Tang et al. (2020)], a publicly available pre-processing tool for electronic health record data. We used the same features as [Tang et al. (2020)] for our tasks. More information can be found at https://physionet.org/content/mimic-eicu-fiddle-feature/1.0.0/.

### B.2 ADULT

Although, we used a pre-processed version of this dataset, we omitted features pertaining to education, work type, and work sector to make the task more difficult. More specifically, in the file 'headers.txt' at the repository mentioned in Footnote 1, we kept all features beginning with 'age', 'workclass', 'education', 'marital status', and 'occupation'. We also kept the 'Sex_Female' feature. Values were normalized for each feature to have a range of 0-1 by subtracting by the minimum value observed among all individuals and dividing by the range.

Table 2: For each dataset, we list the range of hyperparameters considered for each dataset. For each hyperparameter, the lower bound is shown in the top row, and the upper bound is shown in the bottom row. For hyperparameters we did not tune, only one row is shown.

| Hyperparameter | Synthetic | MIMIC-ARF | MIMIC-Shock | Adult | COMPAS |
|---|---|---|---|---|---|
| Learning Rate | 0.00001 | 0.00001 | 0.000001 | 0.00001 | 0.0001 |
| | 0.01 | 0.001 | 0.001 | 0.01 | 0.05 |
| L2 Constant | 0.0001 | 0.000001 | 0.0001 | 0.0001 | 0.0001 |
| | 0.1 | 0.01 | 0.1 | 0.1 | 0.01 |
| Layer Size | 10 | 500 | 500 | 100 | 10 |
| Filter Threshold | 0.40 | 0.50 | 0.50 | 0.50 | 0.50 |
| | 1.00 | 1.00 | 1.00 | 0.90 | 0.90 |
| Noise Added | 0.00001 | 0.00001 | 0.00001 | 0.0001 | 0.0001 |
| | 0.01 | 0.001 | 0.001 | 0.001 | 0.01 |
| Number of Parts | 1 | 1 | 1 | 1 | 1 |
| | 10 | 10 | 10 | 10 | 10 |
| $\alpha_{GPL}$ | 0.01 | 0.1 | 0.001 | 0.01 | 0.01 |
| | 1.0 | 1.0 | 1.0 | 1.0 | 1.0 |
| $\alpha_{1Proposed}$ | 0.1 | 0.1 | 0.01 | 0.1 | 0.01 |
| | 10.0 | 10.0 | 10.0 | 10.0 | 10.0 |
| $\gamma_{Proposed}$ | 0.1 | 0.1 | 0.01 | 0.1 | 0.01 |
| | 10.0 | 10.0 | 10.0 | 10.0 | 10.0 |
| $\alpha_{2Proposed}$ | 0.1 | 0.1 | 0.01 | 0.1 | 0.01 |
| | 10.0 | 10.0 | 10.0 | 10.0 | 10.0 |

### B.3 COMPAS

Although, we used a pre-processed version of this dataset, we omitted the feature 'score_factor' (i.e., the risk score for recidivism from the ProPublica model) to make the task more difficult. Values were normalized for each feature to have a range of 0-1 by subtracting by the minimum value observed among all individuals and dividing by the range.

## C NETWORK AND TRAINING DETAILS

Here, our ranges of hyperparameters and implementation choices for the proposed network. All networks were trained on Intel(R) Xeon(R) CPUs, E7-4850 v3 @ 2.20GHz and Nvidia GeForce GTX 1080 GPUs. All layers were initialized with He initialization from a uniform distribution. We also include are source code as part of the submission in the supplemental attachment.

### C.1 HYPERPARAMETER VALUES CONSIDERED

Here, we show the range of values we considered for our random search. More details are provided in **Table 2**. For any hyperparameters associated with the Adam optimizer not mentioned above, we used the default values. We divide our training data into five batches during training. Fandom seeds (for Pytorch, numpy, and Python's random) were initialized with 123456789.

Not all hyperparameters were used with each approach. 'Filter Threshold' and 'Noise Added' were only used with the baseline SLN + Filter. Here, Filter Threshold refers to the minimum value of the predicted probability of the observed label for an instance to be considered 'correctly labeled'. For example, if Filter Threshold=0.5, then all examples whose predicted probability for the observed label is at least 0.5 are considered 'correct' and used during training. 'Number of Parts' was only used with the baseline Transition. '$\alpha_{GPL}$' was only used with the baseline Fair GPL, and '$\alpha_{Proposed}$' was only used with the proposed method. Here, $\alpha_{Proposed}$ corresponds to the term $\alpha$ that was used to balance $\mathcal{L}_\theta$ and $\mathcal{L}_\phi$. We refer to it as $\alpha_{Proposed}$ in this section to distinguish it from the $\alpha$ value used by the baseline Fair GPL.

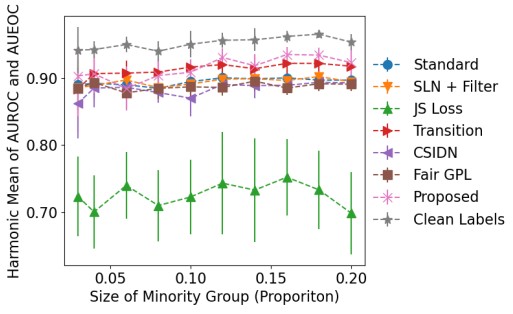 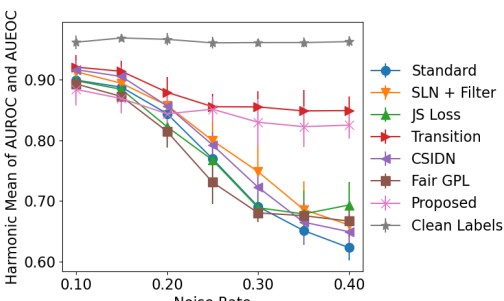

(a) Comparison of the proposed approach to baselines when varying the size of the minority set in the whole dataset. The proposed approach outperforms the baselines on minority sizes of 10% or higher.

(b) Comparison of the proposed approach to baselines with uniform random noise. Overall, the proposed approach is robust to the noise rate and is competitive with the baselines.

Figure 4: Extra experiments on different data and noise settings.

## C.2 NETWORK DETAILS

For the overall architecture, we used a feed forward network with two hidden layers. The auxiliary $\beta$ prediction component was also implemented with two feed forward layers. All layer sizes are as described in **Table 2**. In addition, we used the ReLU activation function. The complete implementation can be found in the attached code.

## C.3 SOURCE CODE

See the attached zip file. We will make the code public upon publication.

## D IMPLEMENTATION DETAILS

For each dataset, we randomly split the data into 80/20% training/test, ensuring that data from the same individual did not appear across splits. For the Adult dataset, we used the test set provided and randomly selected 1,000 individuals from the training set. We then randomly selected 10% of the training data for all datasets except MIMIC-III from each subgroup to be alignment points, thereby ensuring that they were representative of the overall population. For the MIMIC-III dataset, 2% from each subgroup were selected as alignment points due to the larger size of the dataset. Alignment points were selected randomly to simulate our setting of focus, where we have a proxy labeling function and then randomly select a subset of the data to chart review in order to validate the proxy function. Then, for all datasets, half of the alignment points were then set aside as a validation set to use during training for early stopping and hyperparameter selection, while the other half remained in the training set. Later, in our experiments, we evaluate when the alignment set size varies and when the alignment set is biased. **All approaches (i.e., baselines and proposed) were given access to ground truth labels for data in the alignment set so that some approaches did not have an unfair advantage.**

All models were trained in Python3.7 and Pytorch1.7.1 [Paszke et al. (2017)], using Adam [Kingma & Ba (2014)]. Hyperparameters, including the learning rate, L2 regularization constant, and objective function scalars (e.g., $\alpha$), were tuned using random search, with a budget of 20. We used early stopping (patience=10) based on validation set performance, where we aimed to maximize the HM. We report results on the held-out test set, showing the mean and standard deviation over 10 replications.

## E RESULTS: ADDITIONAL EXPERIMENTS (SYNTHETIC DATA)

We examined performance when we varied the size of the minority group (**Figure 4a**) and found that our approach is no longer beneficial at minority groups smaller than 10% of the dataset. This

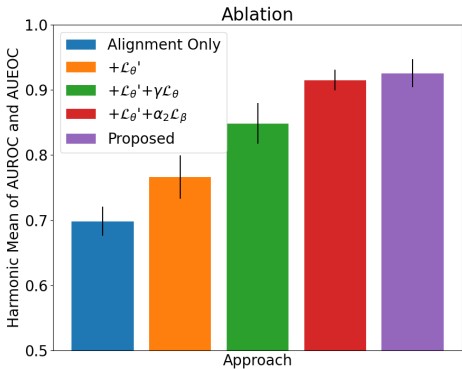

Figure 5: Ablation study of proposed approach.

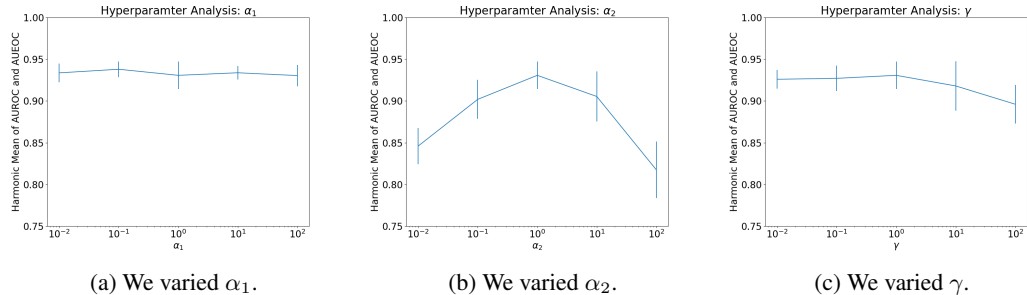

(a) We varied $\alpha_1$.      (b) We varied $\alpha_2$.      (c) We varied $\gamma$.

Figure 6: Sensitivity analysis of proposed approach on objective function hyperparameters.

presents a limitation for future work. We also examined uniform random noise (**Figure 4b**). At low noise rates, all approaches achieved similar performance. At a noise rate of 40%, all approaches outperformed the standard approach, which did not account for label noise. Our approach was still robust to the noise rate and competitive with the baselines. However, since our approach focused on learning the relationship between the features and label noise, we expected that our approach would not provide as large of a benefit on random noise. This is because the only signal our approach would learn is now based solely on the observed label, which may get diluted out if there are many input features.

In addition, we also examine our approach more closely by conducting an ablation study and a hyperparameter sensitivity analysis. In our ablation study (**Figure 5**), we began with training on only the alignment points (i.e., Step 1 only), which achieved the worst performance. We then introduced Step 2 and added the remaining training data (i.e., non-alignment points) but only trained using $\mathcal{L}_{\theta'}$. This led to an improvement in performance, but not to the level of the full approach. The next two ablations build on the previous one. In the first one, we added continued supervision on the alignment points with $\mathcal{L}_{\theta}$, and observed an improvement in performance, likely due to the retention of high quality data in this step. In the second one, we added continued supervision on the alignment points using $\mathcal{L}_{\phi}$, and observed an even larger improvement. This is likely because including $\mathcal{L}_{\theta}$ prevented the model from learning a solution where $\hat{\beta}$ was small for all instances, as previously discussed. Finally, we end with our full proposed approach, which performed noticeably better than each of the ablations, showing the importance of each component.

In our sensitivity analysis (**Figure 6**), we tested how performance of the (full) proposed approach varied to changes in the hyperparameters $\alpha_1$, $\alpha_2$, and $\gamma$. For each of these hyperparameters, we measured performance at values between 0.01 and 100 on a logarithmic scale while keeping the other two values constant at 1. We found that $\alpha_1$ and $\gamma$ were the most robust to changes in the value. We found that $\alpha_2$ was more sensitive, with values between 0.1 and 10 generally working best.

