# OpenReview forum: "Learning with Instance-Dependent Label Noise: Balancing Accuracy and Fairness"
_ICLR.cc/2023/Conference — Submitted to ICLR 2023_

### Official Review · Reviewer_3mWV · 2022-10-23

**Confidence:** 3
**Correctness:** 3
**Technical Novelty And Significance:** 2
**Empirical Novelty And Significance:** 4
**Recommendation:** 3

**Clarity, Quality, Novelty And Reproducibility:**

Quality: ordinary

Clarity: good

Originality: ordinary

**Strength And Weaknesses:**

Strength:
1. They link instance-dependent label noise and algorithmic fairness, which is a practical and significant in real scenarios.

2. Their experiments are thorough enough and achieve relativelt good performance.

Weaknesses:
1. I think that considering the fairness in instance-dependent label noise learning is a key contribution in this paper. However, I cannot find any quantative results to evaluate the fairness of their proposed method and baseline methods. These results serve as the important support for this claim.

2. Their technical novelty is limited. Anchor points have been thoroughly studied in recent years. Maybe I have not realized their unique technical contributions compared with previous literatures, e.g., Xia et al., 2020. This question should be clearly explained.

3. I find their proposed method often fails to outperform Transition, e.g., Figure 2(b), Figure 4(b). Does this phenomenon indicates that their proposed method fails to address the severe label noise?

**Summary Of The Paper:**

This paper proposes a two-stage method to solve instance-dependent label noise learning with anchor points, taking care of robustness and fairness simultaneously.

**Summary Of The Review:**

This paper aims to address instance-dependent label noise learning based on anchor points, which is not very novel for this area. Promisingly, they propose to consider fairness in this setting. Fairness problem is associated with instance-dependent label noise naturally. However, this paper lacks the corresponding quantative results to justify their advantages towards fairness.

---

> ### Author Response · Authors · 2022-11-15
> **Response to weaknesses**
>
> 1. I think that considering the fairness in instance-dependent label noise learning is a key contribution in this paper. However, I cannot find any quantative results to evaluate the fairness of their proposed method and baseline methods. These results serve as the important support for this claim.
>
> Response: We thank the reviewer for the feedback. In our results (Figure 2 and 3), we combined discriminative performance and fairness and plotted the harmonic mean of the AUROC (measure of discriminative performance) and AUEOC (measure of fairness) to assess general performance with respect to both of these aspects.  We clarify this in the updated Section 4.
>
> Section 4, paragraph 1 (page 6)
>
> “We describe the results from experiments with instance-dependent noise. For each plot, we combined discriminative performance and fairness and plotted the HM of the AUROC and AUEOC to assess general performance with respect to both of these aspects.”
>
> 2. Their technical novelty is limited. Anchor points have been thoroughly studied in recent years. Maybe I have not realized their unique technical contributions compared with previous literatures, e.g., Xia et al., 2020. This question should be clearly explained.
>
> Response: The reviewer is correct in that anchor points have been used in the literature. The novelty in our approach lies in how we use the anchor points. We have updated the related work section to clarify. We would also like to let the reviewer know that we have updated our terminology to refer to “anchor points” as “alignment points since the way we define them differs slightly from past work. We clarify this as well.
>
> Section 1, paragraph 3 (page 2)
>
> “We leverage alignment points (i.e., a subset of data where we know the observed \textit{and} ground truth labels) that are representative of the target population. In contrast to anchor points [Xia et al 2019], which generally assume that the ground truth and observed labels match for such points, we consider a novel setting in which the ground truth and observed label do not always match for these points. Such settings arise frequently in healthcare when one uses a proxy function to get observed labels.”
>
> Section 5, last paragraph (page 9)
>
> “Another related setting in past work uses anchor points. To date, anchor points are generally used to learn a transition function [\cite{xia2019anchor, xia2020part, berthon2021confidence}] or for label correction directly [\cite{wu2021learning}]. We use a similar concept, alignment points, to 1) pre-train the model, and 2) predict label correctness. The first part builds from work in semi-supervised learning [\cite{cascante2021curriculum}], which has shown improvements from pre-training on labeled data. The second part is similar to a transition function, but differs in that we use the correctness predictions to re-weight the loss rather than adjust the predictions.”
>
> 3. I find their proposed method often fails to outperform Transition, e.g., Figure 2(b), Figure 4(b). Does this phenomenon indicates that their proposed method fails to address the severe label noise?
>
> Response: While our approach generally outperforms baselines across a variety of settings it does not  provide benefit over existing approaches when the noise is completely random or the signal-to-noise ratio is very low. We have updated the manuscript to emphasize the settings in which our approach would not apply.
>
> Section 4, paragraph 3 (page 7)
>
> “In line with our expectations, the proposed approach was robust to changes in the noise disparity. … Although the baseline Transition also demonstrated robustness to the noise disparity and outperformed the proposed approach at minority noise rates higher than 80%, this setting is unlikely to be realistic.”
>
> Appendix, section E, paragraph 1 (page 18-19)
>
> “Our approach was still robust to the noise rate and competitive with the baselines. However, since our approach focused on learning the relationship between the features and label noise, we expected that our approach would not provide as large of a benefit on random noise. This is because the only signal our approach would learn is now based solely on the observed label, which may get diluted out if there are many input features.”

---

### Official Review · Reviewer_LP3P · 2022-10-24

**Confidence:** 4
**Correctness:** 3
**Technical Novelty And Significance:** 2
**Empirical Novelty And Significance:** 3
**Recommendation:** 3

**Clarity, Quality, Novelty And Reproducibility:**

-The presentation of the paper needs to be improved, please see the detailed comments above.

**Strength And Weaknesses:**

Pros:

-This paper tries to solve a general label corruption problem by considering both the accuracy and fairness performance measures, which is very **ambitious**. In the paper, they provide a heuristic approach and empirically analyze its effectiveness on some datasets with artificial instance-dependent label noise.

Cons:

-First of all, the presentation of the paper is **very unclear** and needs significant improvement. For example, in Section 2 problem setup, the statement “learn a model using dataset $D=${$\textbf{x}^i,y^i$}, $i=1,...,n$, that classifies each instance into one of $c$ classes based on $\textbf{\textbf{x}}$” sounds like it’s transductive learning rather than inductive learning; in the last sentence of this paragraph “using $\theta (x)$ we obtain prediction $\hat{y}$”, how do you obtain the prediction? do you use $\hat{y}=argmax_{i\in [c]} \theta_i(\textbf{x})$ or others, this should be made explicit; in the next paragraph, what does “$x_i>0.5$” mean? In proposed approach, the optimization problem is written as $\min_{\theta’,\phi’} \mathcal{L}_\theta+\alpha_1\mathcal{L}_\phi\cdot \mathcal{L}_\theta$ but what are $\theta’,\phi’$? I hope all authors can carefully proofread the whole draft and **write rigorously**.

-Also, the proposed two-step training approach is complicated and contains many heuristics. It would be very helpful if an algorithm can be provided.

-In mainstream label noise literature, anchor points are defined in the clean data domain, that is, an instance $\textbf{x}$ is an anchor point for the class $i$ if $p(y=i\mid\textbf{x})$ is equal to one or very close to one. However, the anchor point concept seems to be **misused** as it means an instance $\textbf{x}$ that we know both the observed noisy label and its ground-truth label in this paper.

-The proposed method has two stages, pretraining and alternative training. The latter is further based on two objective functions which are combinations of the clean-data loss component and noisy-data loss component. To analyze why such a complicated method works well empirically and how each component contributes to good performance, an **ablation study** removing pretraining and removing each loss component is definitely needed.

Questions:

-The proposed method is a combination of many loss terms, and therefore has many combination parameters such as $\alpha_1, \alpha_2, \gamma$, is the performance of the proposed method sensitive to these parameters? Sensitivity analysis may be necessary.

-In the proposed method, $\Phi$ is the network for predicting the label confidence score, but given its loss $\mathcal{L}_\Phi$ I don’t see how the instance-dependency is modeled and learned. This needs to be clarified, otherwise, the problem setting of this paper should not be instance-dependent label noise, but general label corruption.



**Summary Of The Paper:**

-This paper considers a general label noise problem (which does not assume any noise models). It proposes a fully heuristic approach: they firstly pre-train the classification model with some clean data, and then update the classification model by alternating between minimizing a loss on learning label confidence score and another loss on weighted training using the learned label confidence weights on each instance and a fairness weight on each subgroup. They also provide some favorable experimental results in the paper.

**Summary Of The Review:**

-Overall the paper considers an interesting problem, but the current presentation seems not ready, and more experimental analysis is needed.

---

> ### Author Response · Authors · 2022-11-15
> **Response to questions**
>
> 1. The proposed method is a combination of many loss terms, and therefore has many combination parameters such as α1,α2,γ, is the performance of the proposed method sensitive to these parameters? Sensitivity analysis may be necessary.
>
> Response: Thanks for the suggestion. In a sensitivity analysis, we found that α1 and γ were the most robust to changes in the value. α2 was more sensitive, with values between 0.1 and 10 generally working best. We have added this to the paper.
>
> Appendix, section E, paragraph 3 (page 19)
>
> “In our sensitivity analysis (Figure 6), we tested how performance of the (full) proposed approach varied to changes in the hyperparameters $\alpha_1$, $\alpha_2$, and $\gamma$. For each of these hyperparameters, we measured performance at values between 0.01 and 100 on a logarithmic scale while keeping the other two values constant at 1. We found that $\alpha_1$ and $\gamma$ were the most robust to changes in the value. We found that $\alpha_2$ was more sensitive, with values between 0.1 and 10 generally working best.”
>
> 2. In the proposed method, Φ is the network for predicting the label confidence score, but given its loss LΦ I don’t see how the instance-dependency is modeled and learned. This needs to be clarified, otherwise, the problem setting of this paper should not be instance-dependent label noise, but general label corruption.
>
> Response: Instance dependency is captured through the beta term, which is the output of the phi network. The phi network takes as input the features and observed label and uses those features to predict the confidence in the observed label. This prediction is beta. As a result, phi makes instance-dependent predictions. We have clarified in the manuscript
>
> Section 2, paragraph 4 (page 3)
> “$\hat{\beta}_{\phi}=P(y==\tilde{y} \vert \tilde{y}, \textbf{x})$, an instance-dependent prediction for whether the observed label is correct based on $\textbf{x}$ and $\tilde{y}$.”

---

> ### Author Response · Authors · 2022-11-15
> **Response to strengths and weaknesses (2)**
>
> 2. Also, the proposed two-step training approach is complicated and contains many heuristics. It would be very helpful if an algorithm can be provided.
>
> Response: We added this to the appendix (section A.1, page 15).
>
> 3. In mainstream label noise literature, anchor points are defined in the clean data domain, that is, an instance x is an anchor point for the class i if p(y=i∣x) is equal to one or very close to one. However, the anchor point concept seems to be misused as it means an instance x that we know both the observed noisy label and its ground-truth label in this paper.
>
> Response: In the most general sense, anchor points are points where the ground truth label is known. Most papers use this definition for anchor points and assume that the ground truth and observed labels are the same for such points. In our paper, we called them anchor points because we also use this definition. However we did not assume that the ground truth and observed labels are always the same.This arises commonly in healthcare settings in which we have some proxy function for quickly labeling a set of data and gold standard labels obtained from a clinician manually labeling data. In light of this difference with past work, we have changed our terminology to call them alignment points instead. We clarify this in the updated manuscript.
>
> Section 1, paragraph 3 (page 2)
>
> “We leverage alignment points (i.e., a subset of data where we know the observed \textit{and} ground truth labels) that are representative of the target population. In contrast to anchor points [Xia et al 2019], in which the ground truth and observed labels match, we consider a novel setting in which there may be disagreement between the ground truth and observed labels. Such a setting arises frequently in healthcare. Oftentimes, one labels an entire dataset using a proxy function (obtaining observed labels) but also labels a small subset of the data using manual review (obtaining ground truth labels).”
>
> Section 2, paragraph 2 (page 2)
>
> “For each alignment point, we know both $\tilde{y}$ and $y$. … Alignment points are similar to alignment points [Xia et al 2019], but we do not assume that $\tilde{y}^{(i)} = y^{(i)}$ for these points. ”
>
> 4. The proposed method has two stages, pretraining and alternative training. The latter is further based on two objective functions which are combinations of the clean-data loss component and noisy-data loss component. To analyze why such a complicated method works well empirically and how each component contributes to good performance, an ablation study removing pretraining and removing each loss component is definitely needed.
>
> Response: Thanks for the suggestion. We added this to the manuscript. Interestingly the full approach performed noticeably better than each of the ablations.
>
> Appendix, section E, paragraph 2 (page 19)
>
> “In addition, we also examine our approach more closely by conducting an ablation study and a hyperparameter sensitivity analysis. In our ablation study (Figure 5), we began with training on only the alignment points (i.e., Step 1 only), which achieved the worst performance. We then introduced Step 2 and added the remaining training data (i.e., non-alignment points) but only trained using $L_{theta'}$. This led to an improvement in performance, but not to the level of the full approach. The next two ablations build on the previous one. In the first one, we added continued supervision on the alignment points with $L_{theta}$, and observed an improvement in performance, likely due to the retention of high quality data in this step. In the second one, we added continued supervision on the alignment points using $L_{phi}$, and observed an even larger improvement. This is likely because including $L_{theta}$ prevented the model from learning a solution where $\hat{\beta}$ was small for all instances, as previously discussed. Finally, we end with our full proposed approach, which performed noticeably better than each of the ablations, showing the importance of each component.”

---

> ### Author Response · Authors · 2022-11-15
> **Response to strengths and weaknesses (1)**
>
> 1. First of all, the presentation of the paper is very unclear and needs significant improvement. For example, in Section 2 problem setup, the statement “learn a model using dataset D={xi,yi}, i=1,...,n, that classifies each instance into one of c classes based on x” sounds like it’s transductive learning rather than inductive learning; in the last sentence of this paragraph “using θ(x) we obtain prediction y^”, how do you obtain the prediction? do you use y^=argmaxi∈[c]θi(x) or others, this should be made explicit; in the next paragraph, what does “xi>0.5” mean? In proposed approach, the optimization problem is written as  minθ′,ϕ′Lθ+α1Lϕ⋅Lθ but what are θ′,ϕ′? I hope all authors can carefully proofread the whole draft and write rigorously.
>
> Response: Thank you for pointing out these shortcomings. We have worked to make the notation clearer.
>
>  Section 2, paragraph 2 (page 2)
>
> “We train a model using dataset $D=\{\textbf{x}^{(i)}, \tilde{y}^{(i)}\}_{i=1}^{n}$, where $\textbf{x} \in R^d$ and $y \in \{1, 2, ..., c\}$ to learn a function $f: \textbf{x} \rightarrow y$ that can map unseen instances into one of $c$ classes based on their feature vectors. ”
>
> For the second point, the output of theta is y-hat, which is defined in Table 1 as a c-dimensional vector. We have clarified this in the manuscript.
>
> Section 2, paragraph 2 (page 2)
>
> “We aim to learn model parameters, $\theta$, such that $\theta(\textbf{x})$ represents the predicted class probabilities, (i.e., $\hat{y}$). ”
>
> For the third point, “xi > 0.5” means that the value of the feature vector x and index i is greater than 0.5. Also note that this was a toy example, and an arbitrary heuristic that makes decisions based on a feature being above 0.5 was not part of the approach. We have clarified this in the manuscript.
>
> Table 1, caption (page 2)
>
> “Notation. We summarize our notation, with the notation appearing in the left column and a description in the right column. Superscripts in parentheses represent specific instances (e.g., $\textbf{x}^{(i)}$). Subscripts represents indexes into a vector (e.g., $\textbf{x}_i$)”
>
> Section 2, paragraph 3 (page 2)
>
> “ In the following toy example, suppose $f(\textbf{x})=1$ for true positive instances, and $\tilde{m}(\textbf{x})=1$ if $x_1 > 0.5$ and 0 otherwise.”
>
> For the last point, theta’ and phi’ are the initial model parameters from the theta and phi networks that were mentioned in Figure 1b. The “prime” notation was used to distinguish the initial model parameters from the final parameters.
>
> Section 2, paragraph 5 (page 3)
>
> “$\theta'$ and $\phi'$ are the initial parameters of $\theta$ and $\phi$. $\mathcal{L}_{\theta}$”

---

### Official Review · Reviewer_GGCC · 2022-10-25

**Confidence:** 2
**Correctness:** 3
**Technical Novelty And Significance:** 2
**Empirical Novelty And Significance:** 3
**Recommendation:** 6

**Clarity, Quality, Novelty And Reproducibility:**

Overall, the paper is easy to follow. There are some minor typos but do not impact the understanding. One critical problem with the paper is that it fails to explain the "fairness" problem sufficiently. I think the fairness problem applies only to a limited situation when the noise level in minority groups is larger than that in majority groups. However, from the beginning, the paper seems to treat the fairness problem as a universal problem caused by noisy labels, and the "balance" wording also suggests there should be a tradeoff between accuracy and fairness. I think the paper should improve on this part, to make the research problem precise and sufficiently explained.

The paper has proposed an effective solution to the instance-based noise problem with a strong reliance on sufficient anchor points. The experimental results have demonstrated the performance of the proposal compared to a series of baselines from different perspectives. The technical proposed is under the same principle as previous methods in that learns the correlation between the clean and noisy label and weighs the instances by subgroup prediction, but may be different in detailed implementations.  Overall, the proposed method is not totally novel but shows some solutions could be as effective as others.

The paper has produced pages to explain the experimental details. Although there is no link to shared codes, I do not see any issue in reproducibility.



**Strength And Weaknesses:**

I think the paper considers a combination of two existing problems: fairness in learning from noisy labels, and instance-dependent noise. These two problems have already been considered in existing works. Nevertheless, the paper tries to provide an effective solution to such a combination of problems. From the experimental results, the paper's solution is effective. It is comparable to the Transition method with slightly better performance under some settings. However, from the illustration, the improvement seems marginal and is not significant if considering the standard deviation.

The paper has claimed to solve the "balance between accuracy and fairness". The experimental results have used the harmonic mean of an accuracy metric and a fairness metric to demonstrate the proposed method has achieved the balance. On the other hand, the wording of balance suggests that there is some controversy between the two, such that the current work needs to "balance" them. But in the proposed method, I do not see quite strong evidence that a special technique is introduced for increasing the balance at the sacrificing of the classification performance. The subgroup weighting used in the training phase may be a technique to increase the balance. But I think finding the appropriate subgroup is tricky. And such subgroup techniques have been used in previous works (as shown in Section 5->Label noise and fairness).

Overall, the strength of the paper is that it tries to solve a combination of two important problems in learning from noisy labels. While the combination is straightforward, the resulting problem is an important problem and may draw attention from the research community. From the metric, the performance looks good and shows the effectiveness of the solution from a different technical perspective rather than the transition matrix one.

The weakness of the paper is that despite its claim, I do not think the problem of "fairness" has been clearly explained in the paper, and there are few technical contributions towards the "fairness" end. The assumption on the representativeness of the anchor points seems too strong and I do not think it could be easily satisfied practically. If saying the anchor points are sufficient, and the data satisfies the clustering distribution, then I am afraid a semi-supervised method could solve the problem easily without knowing any noisy labels.

**Summary Of The Paper:**

This paper focuses on the problem that current learning methods which handle specially noisy labels may increase the unfairness in prediction. Such a problem exists because in some data sets the minority group often has more noisy annotations. For such kind of data, the paper proposes a method targeting instance-dependent label noise by assuming a group of anchor points, which is sufficient to represent the whole dataset, is available. The paper then proposes to use two models simultaneously: one does the classification, and another predicts how noisy the given label is. The proposed training phase composes of two steps: one step uses only anchor points with both clean and noisy labels to train the two models, followed by another step that alternatively optimizes the two models and uses the predicted noisy level to weigh the non-anchor data points. Finally, the paper did some experiments showing that in various anchor points situations (the number of them, and the bias of them), the proposed method can achieve a good overall performance in both classification and fairness.

**Summary Of The Review:**

The paper has proposed a solution to an important research problem. The solution is not novel enough, but simple and effective. The concern may be on the assumptions made, and whether a semi-supervised solution can easily solve the problem given such strong assumptions. The clarity on the "fairness" part also needs to be increased (maybe significantly).

---

> ### Author Response · Authors · 2022-11-15
> **Response to overall coments**
>
> Comment: Overall, the paper is easy to follow. There are some minor typos but do not impact the understanding. One critical problem with the paper is that it fails to explain the "fairness" problem sufficiently. I think the fairness problem applies only to a limited situation when the noise level in minority groups is larger than that in majority groups. However, from the beginning, the paper seems to treat the fairness problem as a universal problem caused by noisy labels, and the "balance" wording also suggests there should be a tradeoff between accuracy and fairness. I think the paper should improve on this part, to make the research problem precise and sufficiently explained.
>
> Response: We thank the reviewer for pointing this out. Our examination of fairness was mainly empirical, where we evaluated our approach and the baselines when the difference in noise rates between the groups increased. We focused on the setting where the minority group had a higher noise rate since this is more likely to be the case in real data. We have clarified this in the paper.
>
> Section 3, paragraph 7 (page 5)
>
> “Across datasets, we focused on cases where the noise rate in the minority population was always greater than or equal to that of the `majority' group since this is more likely to occur.”
> Please see our earlier response beginning with “This is a great point. We would like to clarify that we did not use the term balance to indicate that one aspect of performance …” for a discussion on the use of the term balance and the revisions we made in the paper.
>
> Comment: The paper has proposed an effective solution to the instance-based noise problem with a strong reliance on sufficient anchor points. The experimental results have demonstrated the performance of the proposal compared to a series of baselines from different perspectives. The technical proposed is under the same principle as previous methods in that learns the correlation between the clean and noisy label and weighs the instances by subgroup prediction, but may be different in detailed implementations. Overall, the proposed method is not totally novel but shows some solutions could be as effective as others.
>
> Response: Please see our earlier response beginning with “This is a great point. We would like to clarify that we did not use the term balance to indicate that one aspect of performance …” for more discussion on how our work differs from previous work.
>
> Comment: The paper has produced pages to explain the experimental details. Although there is no link to shared codes, I do not see any issue in reproducibility.
>
> Response: We have included implementation details in the appendix of the paper and provided the code as a supplementary file of the submission. We will make the code public upon publication of the paper. We will also add a reproducibility statement to the paper so that this information is easier for readers to find.
>
> Page 10
>
> “Section 3 describes our experimental setup, with additional on implementation details in Sections B, C, and D of the Appendix. The code has been included as a supplemental file and will be publicly released upon publication.”
>
> Comment: The paper has proposed a solution to an important research problem. The solution is not novel enough, but simple and effective. The concern may be on the assumptions made, and whether a semi-supervised solution can easily solve the problem given such strong assumptions. The clarity on the "fairness" part also needs to be increased (maybe significantly).
>
> Response: Please see our response above, where we clarified our remark with respect to the cluster assumption. In summary, we used semi-supervised learning assumptions on an auxiliary label that indicated label correctness rather than the actual class label itself.

---

> ### Author Response · Authors · 2022-11-15
> **Response to strengths and weaknesses (2)**
>
> 2. The paper has claimed to solve the "balance between accuracy and fairness". The experimental results have used the harmonic mean of an accuracy metric and a fairness metric to demonstrate the proposed method has achieved the balance. On the other hand, the wording of balance suggests that there is some controversy between the two, such that the current work needs to "balance" them....
>
> Response: This is a great point. We would like to clarify that we did not use the term balance to indicate that one aspect of performance (e.g., discriminative performance) would need to be sacrificed to achieve another (e.g., fairness). Here, “good balance” would refer to approaches that can achieve high discriminative performance while also being fair, whereas “bad balance” would refer to approaches that can possibly achieve good performance in one aspect, but not the other. While an approach with poor performance in both aspects could technically be considered  balanced, we consider this undesirable since such an approach would not be beneficial in practice. We have revised the manuscript to replace the term “balance” with “maintain” (e.g., we would like to achieve good discriminative performance while maintaining fairness).
>
> Example revision: Abstract (page 1), similar revision were made throughout the paper
>
> “On many tasks, our approach leads to consistent improvements over the state-of-the-art in discriminative performance (AUROC) while maintaining model fairness (area under the equalized odds curve, AUEOC).”
>
> The subgroups of interest to a classification task are problem-dependent (e.g., race may be more relevant in some situations while other sensitive attributes, such as gender, may be more relevant in other situations). We defined the groups in our experiments based on past work. Additionally, the papers by Jiang & Nachum and Wang et al that we reference in the Section 5 (related work) operate in a slightly different setting than ours. Jiang et al assume uniform random label noise, and Wang et al assume that the label is dependent only on the sensitive attribute and not the other features. In our paper, label noise can depend on the sensitive attribute and the other features.
>
> Section 5, second last paragraph (page 9)
>
> “Past work proposed to re-weight instances from subgroups during training where model performance is poorer [Jiang & Nachum 2020] in the random noise setting. Others use peer loss [Liu & Guo 2020] within subgroups [Wang et al 2021] but assume that noise depends only on the sensitive attribute. We also train with a weighted loss, but weights are based on predicted label correctness rather than performance on the observed labels.”
>
> 3. The weakness of the paper is that despite its claim, I do not think the problem of "fairness" has been clearly explained in the paper, and there are few technical contributions towards the "fairness" end. ...
>
> Response: We appreciate the feedback. Although we would ideally like to have an anchor set that is representative of the target population, we empirically demonstrate that the approach still works well when the anchor set is biased (i.e., the proportions at which different groups appear do not match the target population). In Figure 3b, for the majority of cases on most datasets, we observe that our approach is beneficial to the baselines over a wide range of anchor set biases (which we measured by the proportion at which the minority group occurs in the anchor set). Although our approach did not outperform the baselines in all scenarios, it was generally the most robust to different biases in the anchor set, despite our initial assumption that the anchor set was representative. In addition, we would like to clarify that our remark with respect to the cluster assumption was not on the observed class labels themselves, but rather auxiliary labels indicating whether the observed label is correct (i.e., whether it matches the ground truth). In this sense, our dataset is similar to what is used in semi-supervised learning, where the anchor points have this auxiliary label and the rest of the data do not, thus our reliance on the cluster assumption with respect to auxiliary labels on whether the observed label is correct. We have clarified this in the manuscript.
>
> Section 2, second last paragraph (page 4)
>
> “Note that, in order to expect a benefit by including the noisy data in Step 2b, we rely on the cluster assumption [Singh et al 2008] from semi-supervised learning, which broadly states that labeled data fall into clusters and that unlabeled data aid in defining these clusters. In the context of Step 2b, "labeled" and "unlabeled" are analogous to whether we know if the ground truth and observed labels match (i.e., alignment point versus non-alignment point), rather than the actual class labels themselves.”

---

> ### Author Response · Authors · 2022-11-15
> **Response to strengths and weaknesses (1)**
>
> 1. I think the paper considers a combination of two existing problems: fairness in learning from noisy labels, and instance-dependent noise. These two problems have already been considered in existing works. Nevertheless, the paper tries to provide an effective solution to such a combination of problems. From the experimental results, the paper's solution is effective. It is comparable to the Transition method with slightly better performance under some settings. However, from the illustration, the improvement seems marginal and is not significant if considering the standard deviation.
>
> Response: We agree that the Transition baseline performs most closely to the proposed method compared to the other baselines. This is because the Transition baseline was designed for instance-dependent label noise, while the others were not. As a result, it was more robust in the settings we tested than the other baselines. Compared to the proposed approach, our approach generally provides consistent improvement over the Transition baseline, except in cases that represent “extreme” situations in the dataset (e.g., a highly biased anchor set or a noise disparity of 70% between the groups). We acknowledge that the proposed approach is no longer beneficial in these unlikely scenarios but included these results to give readers a more complete picture of when our approach works and when it has limitations.
>
> Section 4, paragraph 2 (page 6)
>
> “As expected, all approaches degrade as the noise rate increases, with the proposed approach experiencing the least degradation up to a majority noise rate of 60%. The baseline Transition was also robust to the noise rate, although overall performance was generally worse than the proposed approach. At a noise rate of 60%, it performed similarly to the proposed approach.
>
> Section 4, paragraph 3 (page 7)
>
> “In line with our expectations, the proposed approach was robust to changes in the noise disparity. … Although the baseline Transition also demonstrated robustness to the noise disparity and outperformed the proposed approach at minority noise rates higher than 80%, this setting is unlikely to be realistic.”

---

### Official Review · Reviewer_wG5f · 2022-10-25

**Confidence:** 4
**Correctness:** 2
**Technical Novelty And Significance:** 2
**Empirical Novelty And Significance:** 2
**Recommendation:** 3

**Clarity, Quality, Novelty And Reproducibility:**

Clarity: This paper is well-written, however, the organization of this paper might need to be improved, for instance, the authors should use some part of the paper to discuss the definition of fairness and its relation to their method.

Novelty: The novelty of this paper is limited, using dual networks to predict the underlying true label of noisy sets and then only using samples that are likely to be clean to train the main model is not novel [1].

Reproducibility: Authors released their implementation code, and after examining the code, the experiments are believed to be reproducible.

[1] Jiang, L., Zhou, Z., Leung, T., Li, L. J., & Fei-Fei, L. (2018, July). Mentornet: Learning data-driven curriculum for very deep neural networks on corrupted labels. In International conference on machine learning (pp. 2304-2313). PMLR.

**Strength And Weaknesses:**

Pros:
1. The question of interest is interesting and vital, the fairness of label noise learning methods is an important yet under-appreciated topic.
2. Author attempts to provide theoretical proof to justify that their method is equivalent to the model trained on the clean domain in the ideal case.
3. Authors provide comprehensive experiment settings, which helps the reviewer understands the detail of the experiments.
4. The empirical performance of the proposed method is significant compared with selected baseline methods.

Cons:
1. The biggest issue of this paper is the authors' assumption on anchor points, where authors assumed that "a subset of data for which we know the observed and ground truth labels" and "is representative of the target population". Generally, in the field of label noise learning, only the first assumption holds true, whereas the assumption that anchor points are representative of the target population is too strong. This problem directly hinders the theoretical soundness of this paper, because, in reality, the learned $\theta$ and $\phi$ are bound to be biased.
2. While the authors claimed that they are trying to study the fairness of label noise learning methods, I fail to see any relevant definition or justification regarding fairness. As a contrastive example, [1] studies the well-defined counterfactually fairness, and [1]'s relation to fairness is clearly discussed.

Minor issues:
1. Authors are suggested to use more commonly used benchmarks such as CIFAR or MNIST.
2. Authors are encouraged to include more recent baseline methods.

[1] Wu, S., Gong, M., Han, B., Liu, Y., & Liu, T. (2022, June). Fair classification with instance-dependent label noise. In Conference on Causal Learning and Reasoning (pp. 927-943). PMLR.

**Summary Of The Paper:**

This paper aims to study a "fair" model for learning with instance-dependent label noise. The authors proposed a simple yet intuitive solution, which is first pre-train a classifier $\theta$ and a discriminator network $\phi$ with anchor points. And then use the trained discriminator networks to discriminate the noisy samples from clean samples, and use the predicted clean samples to train the classifier. Which can be theoretically equivalent to the model trained with clean data.

**Summary Of The Review:**

Overall, while the topic this paper aims to study is non-trivial and promising, the content of this paper suggests only a very limited relation to the fairness of label noise learning methods. Also, since the authors' assumption on anchor points is too strong, I cannot agree with the premises of this paper, nor the conclusion it draws. Therefore, I vote for the rejection of this paper.

---

> ### Author Response · Authors · 2022-11-15
> **Response to the reveiwer's cons**
>
> Cons:
> 1. The biggest issue of this paper is the authors' assumption on anchor points, where authors assumed that "a subset of data for which we know the observed and ground truth labels" and "is representative of the target population". Generally, in the field of label noise learning, only the first assumption holds true, whereas the assumption that anchor points are representative of the target population is too strong. This problem directly hinders the theoretical soundness of this paper, because, in reality, the learned θ and ϕ are bound to be biased.
>
> Response: This is a great point. Although we would ideally like to have an anchor set that is representative of the target population, we empirically demonstrate that the approach still works well when the anchor set is biased. We have updated the manuscript to clarify. We would also like to let the reviewer know that we have updated our terminology to refer to “anchor points” as “alignment points since the way we define them differs slightly from past work. We clarify this as well.
>
> Section 1, paragraph 3 (page 2)
>
> “We leverage alignment points (i.e., a subset of data where we know the observed \textit{and} ground truth labels) that are representative of the target population. In contrast to anchor points [Xia et al 2019], which generally assume that the ground truth and observed labels match for such points, we consider a novel setting in which the ground truth and observed label do not always match for these points. Such settings arise frequently in healthcare when one uses a proxy function to get observed labels.”
>
> Section 4, second last paragraph (page 8)
>
> “We hypothesized that our approach works best when the alignment points are representative of the population. Here, we test how the proposed approach performs when this assumption is violated. We varied the amount of bias in the alignment set by changing the proportion at which the subgroups were present. We kept the size of the alignment set constant at 10\% of the training data (2\% for MIMIC-III on both tasks). We observe that the proposed approach is beneficial over a wide range of conditions, i.e., when the minority proportion is 20\%-80\%.”
>
> 2. While the authors claimed that they are trying to study the fairness of label noise learning methods, I fail to see any relevant definition or justification regarding fairness. As a contrastive example, [1] studies the well-defined counterfactually fairness, and [1]'s relation to fairness is clearly discussed.
>
> Response:
> We focused on a measure of fairness based on  equalized odds, which we incorporated into our experimental setup by taking the average of this measure (or, more specifically, 1 - equalized odds so that higher values are better) with the discriminative performance (measured by AUROC) to assess how well the approaches could balance these two aspects of overall performance. We have updated the manuscript as follows.
>
> Section 2, third last paragraph (page 4)
>
>  “Our approach encourages fairness as measured by equalized odds, by upweighting groups with a higher estimated noise rate so that they are not dominated/ignored compared to groups with a lower estimated noise rate. In doing so, we hypothesize that accuracy and related metrics, such as the false positive rate, will improve, thereby improving equalized odds. We focus on equalized odds since, in the domains of interest, metrics like the true and false positive rates are particularly important. For example, in healthcare, we would like to correctly predict who will and will not develop a condition so that the appropriate treatment may be used.”
>
> Section 4, first paragraph (page 6)
>
> “We describe the results from experiments with instance-dependent noise. For each plot, we combined discriminative performance and fairness and plotted the HM of the AUROC and AUEOC to assess general performance with respect to both of these aspects.”

---

> > ### Comment · Reviewer_wG5f · 2022-11-18
> > **Some unresolved concerns**
> >
> > Thanks for the authors' very detailed response. However, my concerns are not fully not addressed.
> >
> > 1. $$\textbf{Alignment points vs anchor points}$$
> > The authors are trying to use the term "alignment points" instead of "anchor points". However, the authors' understanding of anchor points seems problematic. In the 3rd paragraph of the introduction, the authors stated that: $$\textit{In contrast to anchor points, in which the ground truth and observed labels match, we consider a novel setting in which there may be disagreement between the ground truth and observed labels.}$$
> >
> > To the best of my knowledge, the anchor points are defined as a subset of the training data which we know their ground-truth label $Y$ and noisy label $\tilde{Y}$, where $Y$ and $\tilde{Y}$ do not necessarily match. Can authors point out which part of [1], did Xia et al. propose that the $Y$ and $\tilde{Y}$ must match?
> >
> > So far, based on my understanding, authors just change the term "anchor points" into "alignment points", but lack the justification and proper formulation on how alignment points are different from anchor points. This remains the main issue of the paper.
> >
> > 2. $$\textbf{Empricial performance of biased anchor points}$$
> >
> > The authors argue that the performance of the proposed method is still strong when alignment points are biased. However, this could be attributed to multiple reasons.
> >
> > 1). The anchor(alignment) point size is probably too large, the existing work that assumes maximal anchor point size is [2], which is $O(C\times P)$, where $P$ is the size of the parts (patches) and $C$ is the number of classes. We can see that this assumption is independent of sample size $N$. However, in authors' experiments, the anchor point size is $O(N)$, which makes it linearly increase as sample size increases. Usually, we have $N >> C\times P$.
> >
> > 2). The collection of anchor points is not well justified, unlike existing methods [2], where anhor points are selected based on samples that maximize the noise posterior. The authors assume the anchor points are completely accurate. This is unrealistic especially when the anchor point size is $O(N)$.
> >
> > 3). The biased anchor points are biased in what way? From the description by the authors, I sense this bias is due to the imbalance of representation between majority and minority groups. But, when we refer to the bias of the anchor points, it also means the bias within each specific group. For instance, the anchor points are only sampled from a sub-region of the majority/minority group.
> >
> > 3. $$\textbf{Equalized odds fairness}$$
> >
> > 1). Authors should give a clear definition and formulation of the equalized odds, this is a newly introduced concept in the machine learning community [4], and it should not be assumed to be commonly known by the audience.
> >
> > 2). Authors should justify why the proposed method can still maintain fairness when the anchor points are biased, the proof in A.2 is not valid when the learned $\theta$ and $\phi$ are biased.
> >
> > Overall, I think this paper still has space for improvement, specifically, authors should either clarify the definition of alignment points or consider not relying on training models only based on alignment points.
> >
> > [1] Xia, X., Liu, T., Wang, N., Han, B., Gong, C., Niu, G., & Sugiyama, M. (2019). Are anchor points really indispensable in label-noise learning?. Advances in Neural Information Processing Systems, 32.
> >
> > [2] Xia, X., Liu, T., Han, B., Wang, N., Gong, M., Liu, H., ... & Sugiyama, M. (2020). Part-dependent label noise: Towards instance-dependent label noise. Advances in Neural Information Processing Systems, 33, 7597-7610.
> >
> > [3] Hardt, M., Price, E., & Srebro, N. (2016). Equality of opportunity in supervised learning. Advances in neural information processing systems, 29.

---

> > > ### Author Response · Authors · 2022-11-18
> > > **Response to unresolved concerns (3)**
> > >
> > > 3. Equalized odds fairness
> > >
> > > Comment: Authors should give a clear definition and formulation of the equalized odds, this is a newly introduced concept in the machine learning community [3], and it should not be assumed to be commonly known by the audience.
> > >
> > > Response: We define/cited equalized odds in the evaluation section but agree that we should have cited it sooner when we mentioned it in Section 2.
> > >
> > > Section 2, third last paragraph, page 4
> > >
> > > “Our approach encourages fairness, as measured by equalized odds [Hardt et al 2016], …”
> > >
> > > Comment: Authors should justify why the proposed method can still maintain fairness when the anchor points are biased, the proof in A.2 is not valid when the learned $\theta$ and $\phi$ are biased.
> > >
> > > Response: We hypothesize that the label noise to feature relationship, though likely not identical between groups, can be generalized between groups. For example, if our alignment set is biased to contain more majority instances, the relationship learned between the majority instances’ features and the label noise may be able to transfer to the minority instances to some extent. We provide this hypothesis in our updated manuscript.
> > >
> > > Section 4, second last paragraph, page 8
> > >
> > > “We observe that the proposed approach is beneficial over a wide range of conditions, i.e., when the minority proportion is 20\%-80\% (Figure 3b). We hypothesize that this could be because the learned relationship between the features and noise can generalize across groups to an extent.”
> > >
> > > References
> > >
> > > [1] Xia, X., Liu, T., Wang, N., Han, B., Gong, C., Niu, G., & Sugiyama, M. (2019). Are anchor points really indispensable in label-noise learning?. Advances in Neural Information Processing Systems, 32.
> > >
> > > [2] Xia, X., Liu, T., Han, B., Wang, N., Gong, M., Liu, H., ... & Sugiyama, M. (2020). Part-dependent label noise: Towards instance-dependent label noise. Advances in Neural Information Processing Systems, 33, 7597-7610.
> > >
> > > [3] Hardt, M., Price, E., & Srebro, N. (2016). Equality of opportunity in supervised learning. Advances in neural information processing systems, 29.
> > >
> > > [4] Yichen Wu, Jun Shu, Qi Xie, Qian Zhao, and Deyu Meng. Learning to purify noisy labels via meta soft label corrector. In Proceedings of the AAAI Conference on Artificial Intelligence, volume 35,pp. 10388–10396, 2021.

---

> > > ### Author Response · Authors · 2022-11-18
> > > **Response to unresolved concerns (2)**
> > >
> > > 2. Empirical performance on biased anchor points
> > >
> > > Comment: The anchor(alignment) point size is probably too large, the existing work that assumes maximal anchor point size is [2], which is $O(C\times P)$, where $P$ is the size of the parts (patches) and $C$ is the number of classes. We can see that this assumption is independent of sample size $N$. However, in authors' experiments, the anchor point size is $O(N)$, which makes it linearly increase as sample size increases. Usually, we have $N >> C\times P$.
> > >
> > > Response: This is a great point, and we agree that always assuming that 10% of the dataset has ground truth labels is not practical, especially when the dataset is large. In our experiments, we mainly looked at an anchor set size of 10% of the dataset for all of our datasets except for MIMIC-III. Since we used between about 1,000-4,800 instances for training/validation in each of the other datasets, our alignment set size was always less than 500 examples. For the MIMIC-III dataset, since this dataset was larger (about 15,000 instances), we only selected two percent of the training/validation set (about 12,000 instances) to be alignment points. This resulted in an alignment set size of about 240 instances.
> > >
> > > Appendix D, paragraph 1, page 18
> > >
> > > “We then randomly selected 10% of the training data for all datasets except MIMIC-III from each subgroup to be alignment points, thereby ensuring that they were representative of the overall population. For the MIMIC-III dataset, 2% from each subgroup were selected as alignment points due to the larger size of the dataset.”
> > >
> > > Comment: The collection of anchor points is not well justified, unlike existing methods [2], where anhor points are selected based on samples that maximize the noise posterior. The authors assume the anchor points are completely accurate. This is unrealistic especially when the anchor point size is $O(N)$.
> > >
> > > Response: We focus on a setting in which we have a proxy labeling function (that is potentially noisy) that is used to label large scale datasets, and we have a randomly selected subset of the data that is chart reviewed by an expert. Therefore, in our experiments, we chose our alignment points randomly to simulate this process. We have updated the manuscript to clarify.
> > >
> > > Appendix D, paragraph 1, page 18
> > >
> > > “Alignment points were selected randomly to simulate our setting of focus, where we have a proxy labeling function and then randomly select a subset of the data to chart review in order to validate the proxy function.”
> > >
> > > Comment: The biased anchor points are biased in what way? From the description by the authors, I sense this bias is due to the imbalance of representation between majority and minority groups. But, when we refer to the bias of the anchor points, it also means the bias within each specific group. For instance, the anchor points are only sampled from a sub-region of the majority/minority group.
> > >
> > > Response: The reviewer is correct in that the bias represents the proportions at which the majority and minority groups appear. The reviewer also makes a good point in that this could be more fine grained (e.g., looking at subgroups within the minority group), and this could be addressed with our approach by considering the subgroups within groups as separate groups.

---

> > > > ### Comment · Reviewer_wG5f · 2022-12-06
> > > > **Unaddressed concern on anchor point size**
> > > >
> > > > Authors have made the argument that, when the size of the dataset is small, then assuming that 10% of the dataset belongs to the alignment set is mild, and for the larger dataset, they only assumed 2% of the dataset belongs to the alignment set.
> > > >
> > > > However, this does not change the fact that your alignment point size scales linearly as the size of the dataset increases. I recommend authors find existing works in the field of label noise learning to support the claim that "there exists a subset of reliable samples whose labels are reliable, and the sample size scales as the total sample size increases."
> > > >
> > > > Also, can authors elaborate on "considering the subgroups within groups as separate groups"? If the sampling of alignment points is biased, then how can you separate those samples? If you trained the $\phi$ with a biased subgroup, then it won't be able to distinguish clean and noisy samples for the overall distribution.

---

> > > > > ### Author Response · Authors · 2022-12-12
> > > > > **Response to unadressed concerns**
> > > > >
> > > > > We appreciate the feedback. The size of the subset for which one has ground-truth labels is most likely to vary depending on the application. In many practical scenarios in healthcare, this approach reduces the amount of labor intensive labeling. If the alternative is having clinicians perform manual chart review on every single case, only having to label 10% of a dataset represents significant savings. For the reviewer’s inquiry on whether past work has suggested obtaining an anchor set whose size is proportional to the size of the dataset, this has been suggested in the field of survival analysis/time to event analysis [1], where the time at which an event occurs may be incorrectly recorded in the dataset and we would like to select a subset of the data for expert review. More recent work has also chosen the size of the anchor set as a proportion of the size of the dataset [2].
> > > > >
> > > > > From Hunsberger et al [1], where “reference test measures” refer to ground truth measurements for when an event occurred: “We showed that collecting reference test measurements on only 10% of the sample gives more power than collecting no reference test measurements, even for high sensi-tivity and specificity values.”
> > > > >
> > > > > From Oh et al [2], where researchers call the “anchor set” the “validation subset” in the survival analysis literature: “The validation subset was selected as a simple random sample of 20%, resulting in 373 patients.”
> > > > >
> > > > > [1] Hunsberger S, Albert PS, Dodd L. Analysis of progression-free survival data using a discrete time survival model that incorporates measurements with and without diagnostic error. Clinical Trials. 2010 Dec;7(6):634-42.
> > > > >
> > > > > [2] Oh EJ, Shepherd BE, Lumley T, Shaw PA. Raking and regression calibration: Methods to address bias from correlated covariate and time‐to‐event error. Statistics in medicine. 2021 Feb 10;40(3):631-49.
> > > > >
> > > > >
> > > > > We clarify “considering subgroups within groups as separate groups” through the following toy example. Suppose that in our dataset, we have red and green objects that are either circles and squares. If we define “red” and “green” to be our groups of interest, then all we care about is balancing with respect to color. However, if our groups of interest are further separated by shape (e.g, “red circles”), then an anchor set could still be biased if it contains only squares. In our paper, we assume that the groups of interest are known in advance and that group membership is known for all instances. Within this setting, we explored the effects on training with a biased anchor set and demonstrated the limits of our approach.

---

> > > ### Author Response · Authors · 2022-11-18
> > > **Response to unresolved conerns (1)**
> > >
> > > The reviewer has raised some good points. We respond to them below.
> > >
> > > 1. Alignment points vs anchor points
> > >
> > > Comment: To the best of my knowledge, the anchor points are defined as a subset of the training data which we know their ground-truth label $Y$ and noisy label $\tilde{Y}$, where $Y$ and $\tilde{Y}$ do not necessarily match. Can authors point out which part of [1], did Xia et al. propose that the $Y$ and $\tilde{Y}$ must match?
> > >
> > > Response: While Xia et al. did not state that the ground truth and observed labels must match for anchor points, they implicitly assumed this was the case as they used the observed label as the ground truth label for the anchor points in their experimental setup. Additionally, this has been an assumption incorporated into the experiments of other past work [2,4]. Based on this, another reviewer (lp3p) pointed out that our relaxation of this assumption did not perfectly match the past use of anchor points. In response to this reviewer’s feedback, we clarified the setting and changed the terminology in the updated paper.
> > >
> > > Section 1, paragraph 3, page 2
> > >
> > > “We leverage a set of representative points in which we have access to both the observed and ground truth labels. While past work has used the observed labels as the ground truth labels for anchor points [Xia et al 2019, Wu et al 2021], we consider a different setting in which the ground truth and observed labels do not agree for some points. To make the differentiation clear, we refer to these points as “alignment” points.”
> > >
> > > Comment: So far, based on my understanding, authors just change the term "anchor points" into "alignment points", but lack the justification and proper formulation on how alignment points are different from anchor points. This remains the main issue of the paper.
> > >
> > > Response: Please see our response above. In summary, our setting is more specific in that we assume that the alignment set contains instances where the ground truth and observed labels do not match.

---

> > > > ### Comment · Reviewer_wG5f · 2022-12-06
> > > > **Partially addressed concern on "alignent points vs anchor points"**
> > > >
> > > > Thank the authors for the active discussion. However, I must point out that, if your conclusion is drawn from observing the experiments instead of the original paper, then it might not be completely reliable. There are many reasons to explain why their implementation assumes anchor points' true labels are the same as their noisy labels, based on their method of selecting the anchor points.
> > > >
> > > > More specifically, if anchor points are selected based on confident samples (those that maximize the noisy class posterior) [1], then the true labels and the noisy labels will align generally.
> > > >
> > > > However, if we assume anchor points are known and randomly sampled, then their true labels can be different from their noisy labels, and the probability of this event happening is equal to the noise probability.
> > > >
> > > > Therefore, your statement "anchor points assume their noisy labels and clean labels are the same" is empirically true when we have to estimate anchor points ourselves, but not true under the general settings.
> > > >
> > > > [1] Li, X., Liu, T., Han, B., Niu, G., & Sugiyama, M. (2021, July). Provably end-to-end label-noise learning without anchor points. In International Conference on Machine Learning (pp. 6403-6413). PMLR.

---

> > > > > ### Author Response · Authors · 2022-12-12
> > > > > **Response to  "alignent points vs anchor points"**
> > > > >
> > > > > I think we’re on the same page. We believed our original use of the word ‘anchor’ was appropriate. However, based on feedback from another reviewer, we changed our terminology from ‘anchor’ to ‘alignment.’ In short, we are using anchor points in which the ground truth and and observed labels do not necessarily match (and moreover should *not* perfectly match). To address the earlier reviewer’s confusion, we clarify this in the updated version of the manuscript and use the term ‘alignment’ - but if there’s agreement among the reviewers, we can just as easily revert back to calling these points ‘anchor’ points.

---

> ### Author Response · Authors · 2022-11-15
> **Response to the reviewer's minor issues**
>
> 1. Authors are suggested to use more commonly used benchmarks such as CIFAR or MNIST.
>
> Response: While we acknowledge that these datasets are commonly used in the noisy labels literature, we did not include them in our manuscript since we wanted to evaluate scenarios where harmful biases can arise. We emphasize this now in section 3.
>
> Section 3, second paragraph (pages 4-5)
>
> “While work in noisy-label learning often focuses on image datasets (e.g., MNIST or CIFAR 10/100), given our emphasis on fairness, we selected datasets reflective of tasks in which harmful biases can arise (e.g., predicting clinical outcomes, recidivism, and income).”
>
> 2. Authors are encouraged to include more recent baseline methods.
>
> Response: We thank the reviewer for pointing out the paper by Wu et al. We have included it in our discussion of related work, but do not directly compare to them since we do not have a clear separation of the root level and low level attributes that the paper uses in most of our datasets.
>
> Section 5, second last paragraph (page 9)
>
> “... More recently, Wu et al addressed some of the gaps of past work by examining the instance-dependent case. Our proposed approach differs from theirs in that we do not require our features to be grouped into distinct categories, such as root and low level attributes.”
>
> [1] Wu, S., Gong, M., Han, B., Liu, Y., & Liu, T. (2022, June). Fair classification with instance-dependent label noise. In Conference on Causal Learning and Reasoning (pp. 927-943). PMLR.

---

> ### Author Response · Authors · 2022-11-15
> **Response to the overall comments**
>
> Clarity: This paper is well-written, however, the organization of this paper might need to be improved, for instance, the authors should use some part of the paper to discuss the definition of fairness and its relation to their method.
>
> Response: We have updated Section 2 to more clearly discuss how we define and justify fairness.
>
> Section 2, third last paragraph (page 4).
>
>  “Our approach encourages fairness, which we measure with equalized odds, by upweighting groups with a higher estimated noise rate so that they are not dominated/ignored compared to groups with a lower estimated noise rate. In doing so, we hypothesize that accuracy and related metrics, such as the false positive rate, will improve, thereby improving equalized odds. We focus on equalized odds since, in the settings we focus on, metrics like the true and false positive rates are particularly important. For example, in healthcare, we would like to correctly predict who will and will not develop a condition so that the appropriate treatment may be used.”
>
> Novelty: The novelty of this paper is limited, using dual networks to predict the underlying true label of noisy sets and then only using samples that are likely to be clean to train the main model is not novel.
>
> Response: We clarify the novelty of our approach in Section 5.
>
> Section 5, paragraph 2 (page 8)
>
> “Filtering approaches use heuristics to identify mislabeled instances (e.g., MentorNet [Jiang et al. (2018)], Co-teaching [Han et al. (2018)], FINE [Kim et al. (2021)]). Many are based on the idea that correctly labeled instances are easier to classify than mislabeled instances (i.e., the memorization effect) [Arpit et al. (2017)]. “
>
> Section 5, paragraph 4 (page 9)
>
> “Although we use soft filtering, we filter based on the learned relationship between the features and noise rather than existing heuristics and upweight groups with a higher estimated noise rate.”
>
> Reproducibility: Authors released their implementation code, and after examining the code, the experiments are believed to be reproducible.
>
> Summary Of The Review:
> Overall, while the topic this paper aims to study is non-trivial and promising, the content of this paper suggests only a very limited relation to the fairness of label noise learning methods. ...
>
> Response: While we acknowledge that our setting, which includes anchor points, does not apply to all settings, it is relevant in many fields, especially those where fairness is particularly important. For example, in healthcare, ground truth labels obtained via manual chart review with an expert for a subset of the data are required to validate automated labeling tools. This subset can be derived from a randomly chosen subset of the data, and thus can be constructed so that it is representative of the target population. However, even if the anchor set is biased, we have shown that our approach can generally perform well compared to the baselines. Furthermore, we also show that the size of the anchor set does not need to be large in order for our approach to be beneficial. In addition to the proposed approach, our paper makes a significant contribution by shedding light on this often overlooked setting that frequently arises in healthcare. We have updated the manuscript to clarify. We would also like to let the reviewer know that we have updated our terminology to refer to “anchor points” as “alignment points since the way we define them differs slightly from past work. We clarify this as well.
>
> Section 1, paragraph 3 (page 2)
>
> “We leverage alignment points (i.e., a subset of data where we know the observed \textit{and} ground truth labels) that are representative of the target population. In contrast to anchor points [Xia et al 2019], which generally assume that the ground truth and observed labels match for such points, we consider a novel setting in which the ground truth and observed label do not always match for these points. Such settings arise frequently in healthcare when one uses a proxy function to get observed labels.”
>
> Section 2, paragraph 3 (pages 2-3)
>
> “Although it is not always possible to obtain alignment points, the setting above does occur in practice. For example, in healthcare, automated labeling tools based on the structured contents of the electronic health record are often applied to identify cohorts or outcomes of interest [Upadhyaya et al. (2017); Norton et al. (2019); Tjandra et al. (2020); Yoo et al. (2020)]. However, it is often overlooked that such practical definitions are not always reflective of ground truth, and thus, require validation by comparing to a subset of the data that was chart reviewed by an expert. This subset can be derived from a randomly chosen subset of the data, and thus can be constructed so that it is representative of the target population.”

---

### Author Response · Authors · 2022-11-15
**General comment to reveiwers**

We thank the reviewers for their insightful feedback. We respond to each reviewer individually and have updated our paper.

---

### Decision · Program_Chairs · 2023-01-20

**Decision:**

Reject

**Justification For Why Not Higher Score:**

The technical correctness was not guaranteed and the contents are quite misleading.

**Justification For Why Not Lower Score:**

N/A

**Metareview: Summary, Strengths And Weaknesses:**

The paper studied how to balance the generalization and the fairness under instance-dependent label noise. The technical correctness was not guaranteed and the contents are quite misleading. The problem setting should be made clear, whether there are clean data available for validation or even for training. The definition should be made clear too, and if the name already exists in this area, another name should be chosen for your different concept. Furthermore, the title itself is sufficiently misleading: it suggests that IDN is proposed by the current submission, and the new finding is that by learning under IDN, the accuracy and fairness can be balanced as the result of IDN.